# Deriving Brown Carbon from Multi-Wavelength Absorption Measurements: Method and Application to AERONET and Aethalometer Observations

Xuan Wang[1], Colette L. Heald[1,2], Arthur J. Sedlacek[3], Suzane S. de Sá[4], Scot T. Martin[4], M. Lizabeth Alexander[5], Thomas B. Watson[3], Allison C. Aiken[6], Stephen R. Springston[3] and Paulo Artaxo[7]

[1]Department of Civil and Environmental Engineering, Massachusetts Institute of Technology, Cambridge, MA, USA
[2]Department of Earth, Atmospheric and Planetary Sciences, Massachusetts Institute of Technology, Cambridge, MA, USA
[3]Environmental and Climate Sciences Department, Brookhaven National Laboratory, Upton, NY, USA
[4]School of Engineering and Applied Science, Harvard University, Cambridge, MA, USA
[5]Pacific Northwest National Laboratory, Richard, WA, USA
[6]Earth and Environmental Sciences Division, Los Alamos National Laboratory, Los Alamos, NM, USA
[7]Institute of Physics, University of São Paulo, São Paulo, Brazil

*Correspondence to*: Xuan Wang (xuanw12@mit.edu)

**Abstract.** The radiative impact of organic aerosols (OA) is a large source of uncertainty in estimating the global direct radiative effect (DRE) of aerosols. This radiative impact includes not only light scattering but also light absorption from a subclass of OA referred to as brown carbon (BrC). However the absorption properties of BrC are poorly understood leading to large uncertainties in modelling studies. To obtain observational constraints from measurements, a simple Absorption Ångström Exponent (AAE) method is often used to separate the contribution of BrC absorption from that of black carbon (BC). However, this attribution method is based on assumptions regarding the spectral dependence of BC that are often violated in the ambient atmosphere. Here we develop a new AAE method, which improves upon previous approaches by using the information from the wavelength dependent measurements themselves and by allowing for an atmospherically-relevant range of BC properties, rather than fixing these at a single assumed value. We note that constraints on BC optical properties and mixing state would help further improve this method. We apply this method to multi-wavelength absorption aerosol optical depth (AAOD) measurements at AERONET sites worldwide and surface aerosol absorption measurements at multiple ambient sites. We estimate that BrC globally contributes up to 40% of the seasonally averaged absorption at 440nm. We find that the mass absorption coefficient of OA (OA-MAC) is positively correlated with BC/OA mass ratio. Based on the variability of BC properties and BC/OA emission ratio, we estimate a range of 0.05-1.5 $m^2$/g for OA-MAC at 440nm. Using the combination of AERONET and OMI UV absorption observations we estimate that the $AAE_{388/440nm}$ for BrC is generally ~4 world-wide, with a smaller value in Europe (< 2). Our analyses of observations at two surface sites (Cape Cod, to the southeast of Boston, and the GoAmazon2014/5 T3 site, to the west of Manaus, Brazil) reveal no significant relationship between BrC absorptivity and photochemical aging in urban-influenced conditions. However, the absorption of BrC measured during the biomass burning season near Manaus is found to decrease with photochemical aging with a lifetime of

~1 day. This lifetime is comparable to previous observations within a biomass burning plume but much slower than estimated from laboratory studies. Given the large uncertainties associated with AERONET retrievals of AAOD, the most challenging aspect of our analysis is that an accurate globally distributed multiple-wavelengths aerosol absorption measurement data set is unavailable at present. Thus, achieving a better understanding of the properties, evolution, and impacts of global BrC will rely on the future deployment of accurate multiple-wavelength absorption measurements to which AAE-methods, such as the approach developed here, can be applied.

## 1 Introduction

The radiative impacts of carbonaceous aerosols, which encompass both black carbon (BC) and organic carbon (OC), remain highly uncertain. Aerosol absorption is dominated by BC which is estimated to be the second largest warming agent contributing to climate change in the last Intergovernmental Panel on Climate Change (IPCC) report (IPCC, 2013). However the uncertainty associated with the BC radiative forcing is as large as a factor of two (Bond et al., 2013; Myhre et al., 2013), and recent work shows that the IPCC estimate is likely biased high (Wang et al., 2014). In contrast, OC is typically treated as a purely scattering agent. However, recent studies show that some OC can also absorb light, primarily at UV wavelengths (Arola et al., 2011; Hecobian et al., 2010; Chakrabarty et al., 2010; Kirchstetter et al., 2004; Chen and Bond, 2010). This absorbing OC, so-called brown carbon (BrC), is mainly produced from biomass burning or biofuel combustion (Washenfelder et al., 2015; Ramanathan et al., 2007) but can also be generated from secondary sources involving the photooxidation of anthropogenic and biogenic volatile organic compounds (VOCs) or aqueous-phase chemistry in cloud droplets (Graber et al., 2006; Ervens et al., 2011).

Modeling studies estimate that BrC contributes 20% to 40% of total carbonaceous aerosol absorption and that its absorption direct radiative effect (DRE) ranges from +0.1 to + 0.6 $Wm^{-2}$ (Feng et al., 2013; Lin et al., 2014; Wang et al., 2014; Saleh et al., 2015; Jo et al., 2016). However, all of these studies suffer from substantial uncertainties given that our knowledge of the sources, optical properties, and chemical transformations of BrC are poorly understood. Although we know that BrC is associated with biofuel and biomass burning combustion, the role of fuel source and burning conditions in determining BrC absorption is not well known. Saleh et al. (2014) and Martinsson et al. (2015) suggest that the absorption of BrC generated from biomass burning is correlated with the BC/OC emission ratio. Various secondary organic aerosol (SOA) precursors are also thought to be a source of BrC, including monoterpenes, isoprene, and nitroaromatic compounds (Laskin et al., 2015). However, the formation and abundance of these particles in the atmosphere is poorly constrained. In addition to sources, the uncertainty in estimates of BrC absorption is also driven by uncertainty in optical properties. For example, comparison of measured refractive index (RI) and absorption Ångström exponent (AAE) have been found to differ significantly between studies (Wang et al., 2014; Laskin et al., 2015). Finally, chemical transformations may also alter the optical properties of BrC in the atmosphere. Some studies suggest that absorption increases during the formation and chemical aging of certain types of biogenic and aromatic SOA (Flores et al., 2014; Laskin et al., 2015), while other studies indicate that the absorption

of BrC may decrease during photolysis (Zhong and Jiang, 2011; Lee et al., 2014; Martinsson et al., 2015). Most of these results are from laboratory experiments and require confirmation from field observations. Forrister et al. (2015) use airborne observations of two fire events to show that the absorption associated with BrC decreases following emission, estimating a half-life for biomass burning BrC absorption of 9 – 15 hours. However, this rate is much slower than that suggested by laboratory studies (5 minutes to 3 hours) (Zhong and Jiang, 2011; Lee et al., 2014; Zhao et al., 2015), though none of these has explored how the absorption of primary BrC from biomass burning evolves under oxidizing conditions. Given the above uncertainties, field measurements of BrC are vital not only for constraining models but also for understanding the properties and transformations of this aerosol and its radiative impacts.

To date, the only method for directly measuring BrC absorption involves extraction of filter samples in water, acetone, or methanol. This approach is offline and requires detailed laboratory analysis. This is therefore not a viable approach for obtaining global, continuous measurements; direct observations of BrC absorption from field campaigns are also limited. As a result, more indirect methods based on calculating the difference between total absorption and that of BC have been developed. However, it is clear that the uncertainty associated with separating BrC absorption from BC absorption is larger than the uncertainty associated with the organic extraction method, because the absorption of BC itself can be highly uncertain (Koch et al., 2009; Bond et al., 2013; Wang et al., 2014). To separate BrC absorption from total absorption from satellite or ground-based measurements, one can use complex model retrievals to determine particle type and refractive index (Tesche et al., 2011; Arola et al., 2011). The uncertainty on this approach can be very large and is hard to quantify given that it relies on multiple assumptions regarding aerosol composition and size distributions (Li et al., 2009). Alternatively, because BrC primarily absorbs light in the near-UV, its AAE differs from BC, and therefore a simple AAE method can also be used to estimate BrC absorption:

$$AAE = -\frac{\ln\left(\frac{abs(\lambda_1)}{abs(\lambda_2)}\right)}{\ln\left(\frac{\lambda_1}{\lambda_2}\right)} \tag{1}$$

Here $\lambda_1$ and $\lambda_2$ are two reference wavelengths; abs($\lambda$) is the absorption (or absorption coefficient, or AAOD) at the corresponding wavelength. We can consider the case where we have absorption measurements at three wavelengths, one in the near-UV or a short wavelength including BrC absorption and the other two in the visible spectrum without BrC absorption (for example, 440nm, 675nm and 870nm) to try to separate the absorption of BC and BrC. If there is no dust present, the absorption at the two longer wavelengths (675nm and 870nm) is solely from BC, and the absorption at the shortest wavelength (440nm) includes contributions from both BC and BrC. As a result, if the AAE of BC is known, the absorption of BC at 440nm can be calculated using the longer wavelengths measurements. Then the absorption from BrC can be simply derived by removing this BC contribution to the 440nm measurements. The retrieval of continuous measurements of total aerosol absorption provided by the Aerosol Robotic Network (AERONET) of ground-based sunphotometers since 1992 is an attractive resource for development of such indirect methods. Several studies have applied this idea, using empirically estimated BC-AAE to derive the BrC absorption (Russell et al., 2010; Chung et al., 2012; Bahadur et al., 2012).

In light of the critical need for observational constraints on BrC, in this study we build on previous AAE-based efforts to estimate BrC. We describe a new AAE method to separate BrC and BC absorption and then apply this method to derive BrC AAOD from AERONET observations as well as from a suite of aethalometer field observations of absorption. In doing so, we aim to improve our understanding of BrC emissions and optical properties, as well as provide a new observational constraint for BrC modelling studies.

## 2 Method for deriving BrC absorption from observations

Many studies have applied the simple AAE-based approach to laboratory and field measurements. These studies typically assume AAE = 1 for BC to derive the BrC absorption (Clarke et al., 2007; Herich et al., 2011; Sandradewi et al., 2008; Yang et al., 2009; Liu et al., 2015; Olson et al., 2015 etc.) However, the AAE=1 assumption may not be representative of ambient BC. Lack and Langridge (2013) summarize a series of field measurements and find that AAE of BC (for 467 and 660nm) typically ranges from 0.8 to 1.4. Furthermore, the assumption that BC AAE = 1 under all conditions is also theoretically incorrect. Figure 1 summarizes a series of Mie calculations for single BC particles of varying size and coating. An AAE of 1 is reasonable when the diameter of the BC particle is smaller than 10nm. However, BC associated with biomass burning and biofuel sources is typically larger than 70nm (Bond and Bergstrom, 2006). For these larger BC particles, the AAE is highly sensitive to the size. In addition, coating of BC by other materials, as is commonly observed in the atmosphere (Bond et al., 2013), also modulates the AAE. Finally, the AAE of particles > 20 nm is sensitive to the reference wavelengths chosen. Taken together, the assumption of AAE = 1 for ambient BC is clearly not supported by either theory or field observations, and estimates of BrC absorption based on this underlying assumption are subject to large errors.

Several previous studies have gone beyond the AAE-BC=1 assumption, and used the AAE to separate the BC (or BrC) contribution from total absorption. These analyses typically rely on empirical information from previous observations. For example, Bahadur et al. (2012) and Chung et al. (2012) apply the same approach where they group AERONET sites by regions and possible source types and by analyzing these groups, they estimate the possible AAE (or SSA or EAE) and the corresponding range for pure BC or pure BrC. They then apply these empirical constraints to estimate the BC or BrC contributions at other sites. Bahadur et al. (2012) used this method and found BrC contributes 28% of the total aerosol absorption at 440nm in North America. Chung et al. (2012) concluded that ~20% of the absorption DRE estimated in previous global studies for BC should be attributed to BrC. This approach assumes that the AAE is an intrinsic property by composition, however it is clear from Figure 1, that the AAE is strongly size-dependent, and may therefore vary geographically with combustion conditions. These methods can also only be applied within a given dataset, such as AERONET.

Here we develop a novel method to derive BrC absorption using multiple-wavelength absorption measurements. This AAE-based method, does not rely on assumed or empirically estimated BC and BrC AAE values as in previous studies, but rather combines multiple-wavelength absorption measurements with theoretical Mie calculations for BC. As shown in Figure 1, the

AAE of BC is different when using different reference wavelength pairs. We characterize the WDA (wavelength dependence of absorption angstrom exponent) to describe this difference. This WDA can be seen as the wavelength dependence of the wavelength dependence of absorption, which provides additional information on the aerosol properties that has not been exploited in previous studies. Assuming that we have absorption measurements at 440, 675 and 870nm, then:

$$WDA = AAE_{440/870} - AAE_{675/870} \tag{2}$$

and in the absence of BrC, the $AAE_{440/870}$ and $AAE_{675/870}$ are the AAE of BC calculated using the 440nm/870nm and 675nm/870nm wavelength pairs. Note that the assumption of wavelength-independent AAE=1 for BC would lead to a WDA of 0. For a given population of BC particles, we can use Mie theory to calculate a WDA value by assuming that the particles are spherical. The observed size distribution of BC is typically log-normal, with geometric median diameter (GMD) ranging from 20 to 300 nm and standard deviation ($\delta$) ranging from 1.4 to 2.2 (Akagi et al., 2012; Schwarz et al., 2008; Lack et al., 2012; Dubovik et al., 2002; Shamjad et al., 2012; Moffet and Prather, 2009; Knox et al., 2009; Lewis et al., 2009). We perform Mie calculations using these size distributions and a refractive index of 1.95-0.79i, as suggested by Bond and Bergstrom (2006). We also perform an additional set of calculations for coated BC. The refractive index for coated material is assumed to be 1.55-0.001i, which is the typical value for non-absorbing organic and inorganic (Kopke et al., 1997). We first assume the coating thickness is 10% - 100% of the BC core radius and then only select the calculations with absorption enhancement smaller than a factor of 2. This is supported by field measurements and most laboratory experiments (Schwarz et al., 2008; Lack et al., 2012; Moffet and Prather, 2009; Cappa et al., 2012; Bueno et al., 2011; Shiraiwa et al., 2010; Shamjad et al., 2012; Knox et al., 2009; Liu et al., 2015). Figure 2 shows the range (shaded region) of calculated WDA of BC versus $AAE_{675/870}$ that we estimate based on the above assumptions.

The black line is the median value of the WDA of BC as a function of $AAE_{675/870}$. For a given set of multi-wavelength absorption measurements, if the calculated WDA falls above the shaded region, this suggests that there are components other than BC in the sample which absorb light more strongly at 440nm than at longer wavelengths, supporting the presence of BrC. To illustrate this we take the AAOD measured at all the sites of the global AERONET network as an example. AERONET is a global ground-based aerosol observation network of radiometers (Dubovik and King, 2000; Holben et al., 2001). AERONET AAOD can be calculated at four wavelengths (440, 675, 870 and 1020 nm) based on aerosol optical depth (AOD) and single scattering albedo (SSA), which are retrieved by measuring the sky radiance in a wide angular range. The latest version 2 AERONET product includes two levels of data: 1.5 (cloud screened) and 2 (cloud screened and finally quality-assured). The level-2 AERONET SSA data are only available under high AOD conditions (AOD > 0.4 at 440 nm) (Dubovik and King, 2000; Dubovik et al., 2000); this subset is only 20% of the level-1.5 measurements, which makes the level-2 AAOD biased towards high-aerosol loading conditions. As we want to estimate BrC absorption for a wider range of conditions in the atmosphere, we use the level-2 AOD and SSA as well as recover the missing SSA from level-1.5 in the following analysis. The uncertainty of partially using level-1.5 SSA is hard to estimate, but could be small for our BrC contribution analysis if such uncertainties are similar at all wavelengths.

While AERONET provides global observations of the column-integrated AOD, few of these sites actually have continuous measurements of AAOD throughout the year because the SSA is not always retrieved. For example, more than half of the AERONET sites measured AAOD for only 1 month in 2014. As a result, we use the data from the past decade (2005 – 2014) to enhance our sampling. To reduce the influence of sporadic events in the analysis, when showing the 10 years seasonal

average value only sites with data for more than 6 years within a given season are selected. The AAOD from AERONET not only reflects the absorption from BC and BrC, but also that from dust. We use two thresholds to exclude the data possibly affected by dust. First, we use the coarse-mode AOD contribution (at 440nm) provided by AERONET. We assume that dust controls the total extinction of particles larger than 1 μm diameter (coarse-mode), and therefore remove data with a coarse-mode AOD contribution > 10% from our analysis. Second, we apply the strict filtering of AERONET observations proposed

by Russell et al., 2010 and Chung et al., 2012, excluding data with extinction Ångström exponent (EAE) < 1, as well as Bahadur et al. (2012), excluding data exhibiting scattering Ångström exponent (SAE) < 1.2 and $AAE_{675/870}:AAE_{440/675} < 0.8$. Bahadur et al. (2012) refer to data filtered by this criteria as "dust free". However, we note that AERONET observations are not a direct measurement of absorption, but a retrieved quantity, and though we have attempted to minimize dust contamination in this dataset, retrieval assumptions may also impact our analysis. This is discussed in further detail in

Section 3, but here we apply our methodology to AERONET observations primarily for illustration purposes.

The red crosses in Figure 2 show the calculated WDA using the seasonal average observed AAOD from the global AERONET network in northern hemispheric winter (December, January and February, same sites in Figure 3a). Many points fall within the shaded region, suggesting that for these sites, the absorption at 440nm is primarily from BC. We cannot preclude the presence of BrC in these samples, but the contribution is likely small, and cannot be estimated using our method

without additional information about the size and coating state of BC particles. BrC is clearly present (and contributing to the absorption at 440 nm) for the remaining sites which lie above the shaded region. We calculate the highest and lowest possible BrC absorption at 440nm based on the lowest and highest WDA ($WDA_1$ and $WDA_2$) as follows:

$$BC\ AAE_{440/870} = AAE_{675/870} + WDA \tag{3}$$

$$BrC\ abs(440) = abs(440) - BC\ abs(440) \tag{4}$$

The BrC absorption at 440nm is calculated as the median of these highest and lowest possible absorptions. For those points that fall within the shaded region, the BrC absorption is determined as the median of the highest possible absorption and 0. The methodological uncertainty varies as function of the relative amount of BrC and the measured wavelengths. For example, with measurements of absorption at 440, 675, and 870nm wavelengths, BrC must contribute at least 4% of the total absorption at 440nm to be detected by this approach. This implies a "detection limit" to this approach, where contributions

of 4% or less of BrC to total absorption cannot be identified. This detection limit varies with AAE, and is highest when the AAE of BC is in the range of 1.1 to 1.3. We also estimate the methodological uncertainty range for BrC absorption at 440nm by repeating this calculation using the lowest and highest WDA value of the shaded region ($WDA_1$ and $WDA_2$). For conditions with $AAE_{675/870} > 1$, the methodological uncertainty of derived BrC absorption at 440nm using the above wavelengths is smaller than 28% when the BrC absorption contribution is larger than 30%, but could be as large as 110%

when the BrC contributes around 10% of the total absorption. For conditions where $AAE_{675/870} < 1$, this uncertainty is only 8% when the real BrC absorption contribution is larger than 30% and 35% when the contribution is 10%. Given the modest range in the calculated WDA for BC ($< 25\%$), this method decreases the uncertainty in estimated BrC compared to the traditional BC AAE = 1 method. Lack and Langridge (2013) show that the bias in the traditional BC AAE = 1 method is also

associated with the BrC/BC ratio. The bias from that method is smaller than 33% when BrC contributes 23-41% of total absorption, but much larger (more than 100%) for other BrC contributions. In contrast, for the annual mean observations from the global AERONET network in 2014 that lie above the shaded BC region shown in Figure 2, the uncertainty of BrC absorption derived using our method is smaller than 25%. The spherical assumption in the Mie calculations could lead to additional uncertainties, as previous work suggests that the shape of BC can affect both the SSA and the absorption

enhancement from coating (Adachi et al., 2010; Kahnert and Devasthale, 2011). However this uncertainty is hard to estimate since it is difficult to quantify how particle shape influences AAE and WDA. Our estimated BrC absorption is the externally mixed BrC absorption, which does not include the influence of BrC coated on BC. This is consistent with BrC measurements as the absorption of coated BrC is included in the absorption of BC and cannot be measured separately.

In contrast to previous AAE-based methods, our approach uses the theoretical relationship between AAE and WDA for BC

shown in Figure 2 in combination with the observed total AAE, and does not rely on any other data. This also makes our method "wavelength-flexible". Although we use the 440/675/870nm to describe our method, any three wavelengths with one in the near-UV and two at longer wavelengths in the visible spectrum can be used.

As the absorption from primary OA (Br-POA) from biofuel and biomass burning typically dominates that of absorbing SOA (Br-SOA) (Martinsson et al., 2015; Laskin et al., 2015), the absorption of Br-SOA is much more challenging to detect than

Br-POA in most field measurements. We therefore focus our analysis on the primary sources of BrC.

## 3. AERONET network and data analysis

### 3.1 Global BrC-AAOD from AERONET

Figure 3 shows the derived AERONET BrC-AAOD at 440nm in different seasons. Our BrC-AAOD calculation is based on the daily data from AERONET. This is different from those in Figure 2 (10 years seasonal averaged data, only for

illustration). The AERONET observations of wavelength-dependent absorption are retrieved from the direct and diffuse radiation measured by sun/sky radiometers, but do not include any aerosol assumptions such as those used in the AERONET retrieval of refractive index and size distribution. AERONET AAOD are widely used to investigate the sources, compositions and properties of aerosols (Russell et al., 2010; Bond et al., 2014; Sayer et al., 2014). However we highlight that the retrieval is an indirect measure of aerosol absorption and uncertainties and assumptions in the retrieval scheme may

impact the reported multi-wavelength absorption and introduce subtle inconsistencies with our assumed population of particles. Given the paucity of direct measurements of multi-wavelength absorption (see datasets described in Section 4), we apply our methodology to the AERONET observations to provide a first-look constraint on global BrC AAOD.

The accuracy of specific numerical values presented below is challenging to estimate, we provide these values for completeness in the text, but we focus our conclusions on the qualitative spatial and seasonal differences in estimated BrC which are likely more robust.

For the data points below the methodology detection limit, we calculate the BrC-AAOD as the mean of 0 and the associated detection limit; the mean BrC-AAOD for these points is 0.0034±0.05. The fraction of the data below the detection limit is 22% globally and is regionally consistent. In general, the seasonal average BrC-AAOD is smaller than 0.005 at most sites but larger in Asia. The BrC-AAOD can be as large as 0.056 in the winter at the site near Beijing. The average BrC-AAOD derived at AERONET sites is 0.0031 globally, 0.0018 in North America, 0.0026 in Europe, 0.0119 in East Asia, and 0.004 in South America. The mean BrC-AAOD in the major biomass burning season in Southeast Asia (spring, 0.006) are ~60% higher than non-biomass burning seasons (0.038). In contrast, no significant seasonal variations are found in other regions (data in South America are only available during biomass burning seasons due to the data filtering). The sites in Africa exhibit low BrC-AAOD even during biomass burning seasons. This is because nearly all the data with high AAOD in Africa are excluded from the analysis due to the influence of dust.

Figure 4 shows the contribution of BrC-AAOD to total AAOD at 440nm at each AERONET site. The annual mean BrC AAOD contribution falls below 30% at 80% of AERONET sites. Generally, East Asia and Europe in northern hemispheric winter have higher BrC AAOD contributions (28% and 21%) than other regions/seasons (average 10%). Our estimate in California region (14% in the north and 11% in the south) is comparable with Bahadur et al., 2012 (15% in the north and 9% in the south) during winter and spring, but smaller during summer and fall (~12% vs. ~30%). The OC in both East Asia and Europe have large contributions from residential heating using biofuel, which suggests such biofuel emissions of OC may be more absorbing than other sources that dominate in other seasons. In contrast, in Southeast Asia, the contributions of BrC to total AAOD are relatively aseasonal (< 5% seasonal differences). As the absolute value of BrC-AAOD is larger in the fire season but the contribution of BrC to total absorption is not, the BrC absorption associated with biomass burning from large scale fires may not be very different from the other sources that dominate in other seasons.

The uncertainty on our derived BrC-AAOD (described in Section 2) is different at each site. More than 90% of all the daily data in these 10 years have a methodological uncertainty smaller than 30%. The methodological uncertainty at a given site at a given hour can be as high as 100%. However, very low BrC-AAOD values are derived for these highly uncertain data, having little impact on the 10 year averaged BrC-AAOD. Since AAE < 1 is frequently (60% of the observations for the wavelength pair of 675/870nm) observed at most AERONET sites, substantial BrC absorption would be misattributed to BC using the simple BC-AAE=1 method. By assuming BC-AAE = 1, the global mean BrC-AAOD at 440nm would be estimated as only 0.001, 40% lower than our estimate.

One challenge of this analysis is the well-known uncertainties associated with the AERONET observations. The measurement uncertainty is ±0.01 for AOD, ±0.03 for SSA when AOD > 0.2 and could be as large as ±0.07 for SSA when AOD < 0.2. The uncertainty of AAOD depends on the corresponding AOD value, for example, this uncertainty is ±0.015 with AOD = 0.4 (Dubovik et al., 2002). Because our method is sensitive to the AAE not the AAOD, the uncertainty could be

small for our BrC contribution analysis if such uncertainties from AERONET are similar at all wavelengths. If the AERONET AAOD uncertainties vary substantially with wavelength, the influence on our analysis could be large and hard to quantify. In addition, the AERONET uncertainties suggest AAOD < 0.01 is certainly below the observed detection limit. In Figure 3 and in the above discussion most sites exhibit derived BrC-AAOD smaller than 0.01. However, all of these values

are seasonal means over 10 years, and include both non-BrC detected (BrC-AAOD~0) data and BrC detected data. If instead we replace the BrC-AAOD < 0.01 data points by BrC-AAOD = 0.005 (the median of 0 and 0.01), the results are very similar.

In our method, we assume that the BrC absorption contribution is negligible at 675nm and 870nm. This is supported by the laboratory measurements (Chen and Bond, 2010; Zhang et al., 2013; Yang et al., 2009; Kirchstetter et al., 2004). However,

Alexander et al. (2008) find the BrC absorption may be significant at 675nm by examining an electron energy-loss spectrum from a transmission electron microscope. If BrC absorbs significantly at 675nm, our estimate of BrC absorption at 440nm would be underestimated.

## 3.2 Relationship between BrC-AAOD and BC-AAOD

Figure 5 compares the derived BrC-AAOD at 440nm and BC-AAOD at 675nm at AERONET sites for 2005-2014. As BrC absorbs little light at wavelengths longer than 600nm, BC-AAOD is effectively equivalent to total-AAOD at 675nm. Globally, the BrC-AAOD and BC-AAOD are moderately well correlated ($R^2$ ~ 0.6, not shown on Figure 5). To identify whether the correlation is different under different conditions we further disaggregate the data by emission type and region. We use the anthropogenic emission inventory of Bond et al., (2007) and biomass burning emission inventory from GFED4

(Giglio et al., 2013) to identify the dominant emission type for each data point. For a given month, if the biofuel or biomass burning emissions of both BC and OC contribute more than 60% of the total emissions in a 2°×2° area around a given AERONET site, we identify the corresponding data points as dominated by that source (70% of data points that do not meet this criteria and are thus excluded). The results are summarized in Figure 5. After the data is separated by dominant source, the correlation between BrC-AAOD and BC-AAOD increases in all regions except the biomass burning dominated

European sites. The correlation slope of BrC-AAOD/BC-AAOD varies by region but is similar for different sources in a same region. Although we select the data using emissions in the surrounding 2°×2° area to denote the regional influence, this separation may be inaccurate if long range transport is a significant source of carbonaceous aerosol at a given site. In addition, this data separation is not able to account for the variability in combustion fuel and conditions. In East Asia, the correlation slope (m) between BrC-AAOD and BC-AAOD is ~0.9. It decreases to ~0.5 other regions.

$$BrC_{AAOD} = m \cdot BC_{AAOD} \tag{5}$$

Equation 5 provides a simple method to estimate the absorption of BrC by measuring BC absorption. On average, globally BrC-AAOD at 440 nm is roughly 50% (*m* ~ 0.5) of the BC-AAOD at 675nm based on global AERONET data.

The AAOD is the total column absorption of aerosols, which can be written as the product of column aerosol mass and the mass absorption coefficient (MAC) of aerosols. Based on the linear relationship between BrC-AAOD and BC-AAOD, we are able to connect the MAC of OC (identical to the MAC of BrC when assuming all OC are BrC) with the mass ratio of BC/OC:

$$MAC_{OC} = m \cdot MAC_{BC} \cdot \frac{Mass_{BC}}{Mass_{OC}} \tag{6}$$

The MAC of OC is related to the properties of OC such as size distribution, mixing state, hygroscopic growth and refractive index (RI). Generally, there is a positive correlation between the MAC and the imaginary part of the RI ($i$) although the relationship is not linear (Bond and Bergstrom, 2006). This tells us that the $i$ of BrC is likely to be positively correlated with the mass ratio of BC/OC in a certain environment, as shown by Saleh et al., (2014). The observed relationships shown in Figure 5 alongside of equation 6 confirms that the absorption properties of BrC are likely related to the emission ratio of BC/OC, which further connects to fuel types and combustion conditions. The absorptivity of OC emitted from sources with higher BC/OC is likely to be higher.

Both the BC optical properties and the BC/OC ratio may vary under different conditions. It is therefore challenging to estimate the MAC of OC accurately based on equation 6. However, we can estimate typical values of regional average MAC of OC given that the regional BC/OC emission ratios do not vary substantially in emission inventories, and assuming that the emission ratio of BC/OC is a reasonable proxy for the mass concentration ratio (i.e. relative differences in losses and sources are negligible). Based on the biofuel emissions of Bond et al. (2007), the BC/OC emission ratio is $0.18 \pm 0.03$ in North America and Europe, and $0.24 \pm 0.06$ in other regions. The BC/OC emission ratio for biomass burning in the GFED4 inventory is $0.12 \pm 0.06$ for different source types. Considering the typical size distributions and coating thickness for BC from biofuel and biomass burning sources, the MAC of BC from these sources is calculated to be $7.8 \pm 5$ m$^2$/g by Mie theory (Wang et al., 2014). We therefore estimate the average MAC of OC to be 0.7 m$^2$/g from biofuel in North America/Europe, 0.94 m$^2$/g from biofuel in other regions and 0.47 m$^2$/g from biomass burning at 440nm. Taking the mass ratio of total organic aerosol (OA) and OC to be 2.1 (Turpin and Lim, 2001; Aiken et al., 2008), the corresponding MAC for OA are 0.33, 0.45 and 0.22 m$^2$/g. Considering the variability in the correlation slopes in Figure 5, the BC/OC emission ratios, the size distribution and the mixing state of BC, we estimate a range of MAC of 0.1-3.1 m$^2$/g for OC and 0.05-1.5 m$^2$/g for OA at 440nm. The upper limit is comparable to the highest MAC of acetone/methanol-soluble OA found in laboratory experiments (Kirchstetter et al., 2014; Yang et al., 2009). We note that these estimates of OC-MAC (or OA-MAC) are subject to uncertainties associated with the AERONET retrieval of absorption. It should be mentioned that the MAC of OA reflects both the BrC-MAC and the BrC contribution to the total OA. Since we cannot isolate the BrC contribution to OA, we use the term OA-MAC instead of BrC-MAC to characterize the absorption of OA in this and following analysis.

### 3.3 BrC-AAE from combining AERONET and OMAERUV data

From AERONET data, we can derive BrC-AAOD at one wavelength only (440nm). However, this is insufficient for estimating the full radiative impacts of BrC. To estimate the AAE of BrC, AAOD at least one more near-UV wavelength is needed. Here we use the OMAERUV (near-UV Ozone Monitoring Instrument aerosol algorithm) (Torres et al., 2007; 2014) product together with AERONET to calculate the AAE of BrC.

OMI is a nadir-viewing spectrometer aboard NASA's Earth Observing System's (EOS) Aura satellite. The Aura polar-orbiting satellite orbits with a 16-day repeat cycle and a local equator crossing time of 13:45 ± 15 minutes. OMI measures near-UV radiance at 354 and 388nm and reports AOD, SSA and AAOD at 354, 388, 500nm with a spatial resolution of $13 \times 24$ km$^2$. The AAOD at 388nm is directly retrieved from radiance absorption while the other two wavelengths are derived from 388nm data. In the analyses below, we use the AAOD at 388nm from the level 3 OMAERUVd gridded product and only select the highest quality data by filtering out retrievals affected by large solar zenith angle (>70°), out-of-bounds AOD (>6 at 500nm) or SSA (>1), low terrain pressure (<628.7 hPa), cloud contamination, and cross track anomaly. The root mean square error of the AAOD is estimated to be ~0.01 (Torres et al., 2007).

Given that the measurement method of AERONET and OMAERUV are very different and the temporal coverage of both are relatively poor, we do not combine the OMAERUV AAOD at 388nm with the AERONET AAOD at other wavelengths to calculate BrC-AAOD at 388nm for each AERONET site. Instead, we statistically compare AAOD observations in different regions between these two products. We assume that the distribution of AAOD from a large group of data points should be similar between AERONET and OMAERUV despite differences in the measurement approach. In Figure 6, the frequency distributions for AAOD of OMAERUV (388nm) and AERONET (440nm) are plotted as solid black and red lines. To compare the observations at the same wavelength, we transfer the AERONET AAOD to 388nm by fixing the $AAE_{388/440nm}$ of BC and BrC. When assuming $AAE_{388/440nm} = 1$ for BC, the 388nm AERONET AAOD distribution with different BrC-$AAE_{388/440nm}$ are shown as dashed lines in Figure 6. Assuming a different BC-$AAE_{388/440nm}$ in the range of 0.5 to 1.5 only slightly alters the dashed lines with BrC-$AAE_{388/440nm} = 2$, other dashed lines are largely unaffected. This arises due to the dominance of BrC on total AAE when BrC-$AAE_{388/440nm} > 2$.

In Figure 6a, it is clear that the 388nm AERONET AAOD distribution with BrC-$AAE_{388/440nm}$ =4 (orange dashed line) is the best match to the OMAERUV AAOD (black solid line). This suggests the global mean AAE of BrC should be ~ 4. Similar results are found in North America (Figure 6b), East Asia (Figure 6c) and the rest of the world except Europe. In Europe, regardless of our choice of BrC-$AAE_{388/440nm}$, the 388nm AERONET AAOD distribution does not match the OMAERUV AAOD. This suggests that both BC and BrC may have smaller AAE in Europe. As previously mentioned, when BrC-$AAE_{388/440nm}$ is smaller than 2, the 388nm AERONET AAOD distribution is sensitive to not only BrC-$AAE_{388/440nm}$ but also to the BC-AAE. Thus, by combining OMAERUV and AERONET AAOD data, we find that BrC-$AAE_{388/440nm}$ is typically~ 4 globally but smaller (<2) in Europe. These values are smaller than laboratory measurements of fresh emission from pyrolyzing wood by Chen and Bond, 2010 ($AAE_{380/460nm} > 7$) and biomass burning smoke by Kirchsteter et al., (2004)

(AAE$_{350/450nm}$ = 4.8). It should be mentioned that these AAE values are based on 388-440nm pair and may change for other wavelengths pairs.

Many studies have evaluated OMAERUV AOD by comparing them with ground-based measurements. The correlation between OMAERUV and AERONET AOD is usually found to be high (R > 0.8) (Jethva and Torres, 2011; Ahn et al., 2014). Jethva et al. (2014) also compare the SSA between OMAERUV and AERONET and find 69% of the data agree within the absolute difference of ±0.05 for all aerosol types. Significant differences between the two datasets are most shown at dust dominated sites. These dust-influenced sites are not included in our analysis. Furthermore, Jethva et al. (2014) compare these products at 440nm. The OMAERUV SSA estimated at this wavelength relies on a number of assumptions and is more uncertain than that reported at 388nm that we use in our analysis. It is not possible to directly compare the SSA/AAOD at 388nm since AERONET does not make measurements at this wavelength. Therefore we believe that the comparison between AERONET and OMAERUV is still valid. . However, if the OMAERUV SSA is higher or lower than AERONET at 388nm, our estimate of the BrC-AAE$_{388/440nm}$ would be biased low or high.

## 4. Surface multiple-wavelength absorption measurements

### 4.1 Sites and instruments

In addition to the retrieved column AAOD from AERONET, we also use our method to derive the BrC absorption from direct measurements of absorption at a series of surface sites. Among the measurement methods for aerosol absorption, only the filter-based method using an Aethalometer (AE, Magee Scientific, http://www.mageesci.com) can be used to derive BrC absorption with our method. Aethalometers are designed to measure BC mass concentrations and can measure aerosol absorption at 7 wavelengths ranging from 370nm to 950nm (version AE-31). These 7 wavelengths include more than 2 wavelengths at both UV and long wavelengths (>600nm). As described in Section 2, our method requires absorption measurements in at least 1 wavelength in the near-UV and another 2 measurements at wavelengths >600nm. None of the other commercial instruments currently available meet this requirement.

As a filter-based method, Aethalometer measurements are known to exhibit artefacts from filter loading, filter scattering and aerosol multiple scattering (Liousse et al., 1993; Collaud Coen et al., 2010). It is commonly thought that the absorption measurements from Aethalometers are biased towards much higher values (Arnott et al., 2005; Schmid et al., 2006). Although several correction algorithms have been published, many of these require additional information. Different correction methods may even lead to very different corrected results for the same original data (Schmid et al., 2006; Arnott et al., 2005; Collaud Coen et al., 2010; Weingartner et al., 2003). In our analyses of surface measurements, we will focus on the BrC absorption contribution and BrC-AAE, which are both ratios of AAOD. Following Section 2, the measurement bias on the absolute absorption will have minimal impact on the ratio unless there is a wavelength-dependent bias in the uncorrected data. By analyzing a series of different correction algorithms, Collaud Coen et al. (2010) conclude that it is not possible to precisely estimate the expected bias in AAE, but that the corrected AAE is most likely to remain the same or

increase slightly. In our BrC derivation method, a small change in AAE will not significantly impact the estimated WDA or the BrC contribution. Thus, we can apply our method to derive the BrC absorption contributions from the uncorrected multiple wavelength absorption measured by Aethalometer. We also use these measurements to analyze the variation of BrC absorption at a single site though the absolute values are likely biased high. The Aethalometer data is uncorrected in the following analyses except where noted.

Figure 7 shows the locations of the 10 surface sites we use in this analysis. They are Zeppelin Mountain in the Arctic, Barrow in Alaska, Tiksi in northern Siberia (on the shore of the Laptev Sea), Cool (near Sacramento in northern California) and Cape Cod (to the southeast of Boston) in the United States, Ispra in northern Italy, Preila in eastern Lithuania, SIRTA southwest of Paris in France, Finokalia in Greece (on the northeastern coast of the island of Crete) and the T3 site of the Observations and Modeling of the Green Ocean Amazon campaign (GoAmazon2014/5) to the west of Manaus in Brazil. Detailed information on these sites including measurement periods and references are given in Table 1. The absorption data from the GoAmazon-T3 site is provided with correction for filter loading and multiple scattering effects using the methods outlined by Rizzo et al. (2011) and Schmid et al. (2006).

## 4.2 Estimated BrC absorption and AAE

At Zeppelin Mountain and Cool, we detect little to no BrC absorption with our method. The Zeppelin Mountain site is very clean and aerosol absorption here is generally associated with long range transport. The Cool site is located east of Sacramento and as there were negligible biofuel or biomass burning emissions in July of 2010, no major BrC sources are likely to impact this site. Br-SOA may contribute to the absorption at Cool, but it is below the detection limits of our method. Figure 8 shows the monthly variation in the contribution of BrC to total aerosol absorption at 370nm at the 8 other sites. Since these 8 sites are located in different environments from rural to urban, the chemical composition of OC is likely to be very different among these sites. However, the monthly mean contributions of BrC absorption at 370nm occupy a relatively small range from 7% to 35%. These numbers are smaller than the near-surface values (around 50% at 365nm) estimated for a fire plume based on filter extracts from airborne measurements during the Deep Convective Clouds and Chemistry (DC3) campaign (Liu et al., 2015). It is possible that the BrC contribution is higher in such a concentrated plume compared to the well-mixed air represented by our monthly averages. In addition, while BrC absorption was measured directly during DC3, their estimate of the contribution of BC to total absorption was based on the (possibly inaccurate) assumption that BC-AAE = 1. The annual average total aerosol absorption at the urban site Ispra is nearly 120 times larger than the background site of Barrow in Alaska, however BrC contributes comparable amounts to the total absorption at these sites. In Section 3.1, we showed similar BrC contributions of <30% (at 440 nm) at 80% of AERONET sites; the similar results obtained here from direct measurements of absorption support the analysis of the AERONET data despite concerns about data quality and retrieval assumptions. It is likely that the proportion of BC absorption and BrC absorption do not differ substantially among regions even though the emission sources are very different. This is consistent with our speculation in Section 3.2 that the MAC of OC is related to the combustion properties and positively correlated with BC/OC emission ratio. Because higher

OC-MAC is usually associated with higher BC/OC mass ratio, the absorption ratio of BC/OC may be roughly constant if BC-MAC is relatively constant. This ensures that the proportion of BC to BrC absorption does not vary much among different sources. Small seasonal variations in the fractional contribution of BrC absorption are seen in the sites of Finokalia and SIRTA (Figure 8b and 8c). At these two sites, BrC contributes more absorption in winter than in summer. This is also similar to the analysis of the AERONET observations, however these variations are not large and the monthly mean BrC absorption contributions remain in the 10%-30% range. This winter shift is likely to be associated with the residential heating from biofuel combustion. The BC/OC ratio for biofuel emissions from residential heating is typically lower than other sources (Street et al., 2004). However, given the variability in BC-MAC it is not clear whether this lower BC/OC mass ratio is the dominant reason for the winter shift.

The contribution of BrC absorption changes with wavelength. Considering the influence of the AAE of both BC and BrC, the contribution of BrC at 370nm should be a little larger than at 440nm. For example, with BrC-AAE = 4, a 20% BrC absorption contribution at 370nm will decrease to 13% at 440nm and 7% at 550nm. By comparing the results from these 8 sites with the nearest AERONET sites, we find that the BrC absorption contributions at 370nm at the surface are similar to the column BrC-AAOD contributions at 440nm from AERONET. This may suggest that the BrC absorption contribution increases with altitude. This vertical difference was also identified in analysis of the DC3 airborne observations (Liu et al., 2015). Alternatively, it may suggest a high bias in the BrC estimated from AERONET observations.

Figure 8 also shows the estimated $AAE_{370/430nm}$ of BrC from Aethalometer observations. Generally, the monthly mean BrC-$AAE_{370/430nm}$ at these 8 sites ranges from 2 to 4. Low BrC-AAE (< 2) are frequently observed at the three European sites of SIRTA, Prelia and Ispra (Figure 8c, d and e), consistent with the analysis of AERONET and OMI measurements over Europe. At most of the sites (Figure 8a, b, c, f and h), the monthly variations of BrC-$AAE_{370/430nm}$ are similar to the variation in the BrC absorption contributions. As described above, an increase of BrC absorption contribution may be due to either an increase of OC-MAC or a decrease in the BC/OC mass ratio. This suggests the BrC-$AAE_{370/430nm}$ is either positively correlated with OC-MAC or negatively correlated with BC/OC emission ratio. The later case was observed in the laboratory study of Saleh et al. (2014). However, the correlations between BrC-AAE and BrC absorption contributions are only slight to moderate ($R^2 < 0.3$) at these sites.

## 4.3 Estimating the Evolution of OA-MAC

To better understand the absorption properties of BrC and eliminate the influence of aerosol mass, analysis of both BrC absorption and OA mass are necessary. Additional measurements of OA mass are available at 2 of our 8 sites (Cape Cod and GoAmazon-T3).

At the Cape Cod site during the TCAP campaign, the refractory black carbon (rBC) and OA mass were measured using a single particle soot photometer (SP2, Schwarz et al., 2008) and a high resolution time-of-flight aerosol mass spectrometer (HR-ToF-AMS, Canagaratna et al., 2007) in February of 2013. With both derived BrC absorption and measured OA mass, OA-MAC can be directly estimated. As discussed in Section 4.1, the OA-MAC derived from uncorrected Aethalometer data

is biased high, but the relative variation can be used in the analysis. These data are used to examine the influence of emission properties and chemical processing on the absorption properties of BrC. To identify the impact of emission properties on OA-MAC, we compare OA-MAC with the co-measured BC/OC mass ratio (Figure 9a). These are highly correlated ($R^2$ = 0.75), which further confirms the previously discussed relationship between OA-MAC and BC/OC ratio.

The observations at Cape Cod also provide the opportunity to identify whether there is any relationship between BrC absorption and chemical processing during transport. In laboratory studies, both increases of OA-MAC from new BrC-OA generation and decreases of OA-MAC from BrC-OA photolysis (bleaching) have been observed (Laskin et al., 2015; Lee et al., 2014; Flories et al., 2014). The only previous field observations of chemical processing of BrC by Forrister et al. (2015) found that photolysis decreases the MAC of biomass burning sourced BrC, but the rate of change is much slower than
previously estimated from laboratory experiments. Here we use the quantity of *-log(NO$_x$/NO$_y$)* as a photochemical clock. Assuming the photochemical rate of converting $NO_x$ ($NO+NO_2$) to total reactive nitrogen ($NO_y$) is equal to the reaction between $NO_2$ and OH, this photochemical clock can be seen as a measure of chemical processing since emission (Kleinman et al., 2008). The OA-MAC is plotted versus this photochemical clock in Figure 9b, colored by wind direction. The data points from TCAP in February can be divided into 2 groups by wind direction, one is those from the northwest (red points in
Figure 9b), which are mostly affected by transported urban air from Boston; the remaining data which represent background air masses are shown as a second group (blue points in Figure 9b). The OA-MAC, as well as the mass concentrations of both BC and OA, for the data from the northwest are generally larger than the other group due to the influence of fresher urban emissions. We can see from Figure 9b that this source difference dominates the variation of OA-MAC. For either individual group, we find no significant trend with photochemical clock. Our analysis shows that the OA-MAC neither increases nor
decreases with increasing aging time in background or urban air masses. This suggests that either the generation and photolysis of BrC counteract each other or that the influence from chemical processing is much smaller than emissions and transport for these observations. However we note that these wintertime measurements are not optimal for identifying any photochemical processing; additional measurements in multiple seasons are required.

We repeat the analysis applied at the Cape Cod site to the observations from the GoAmazon-T3 site in the Amazon. Mass
concentrations of BC and OA were measured at this site using an SP2 and an HR-ToF-AMS (de Sá et al., personal communication, manuscript in preparation) during two Intensive Operating Periods (IOPs). The IOP1 took place in the wet season, from February 1 to March 31, and the IOP2 occurred during the dry season from August 15 to October 15, both in 2014. Unfortunately, data availability for BC and $NO_x$ concentrations are poor during IOP1. We therefore supplement our analysis with Aerosol Chemical Speciation Monitoring (ACSM) measurements of OA from January-March of 2015. During
this time period the sources of OA at the T3 site are generally from the city of Manaus and the forested region around it, with very low incidence of fires. IOP2 occurs during the biomass burning season and the T3 site is highly influenced by fires during this period (Martin et al., 2015). Figure 10a shows that the measured OA-MAC and BC/OC ratio during IOP2 are correlated ($R^2$ = 0.5) at this site as well.

Figures 10b and 10c compare the measured OA-MAC with the photochemical clock $-log(NO_x/NO_y)$ at the GoAmazon-T3 site under low biomass burning (January-March 2015) and high biomass burning (IOP2) influence. During the low biomass burning season, we select the daytime data from Manaus only (based on wind direction from the east) to eliminate the potential contamination of the photochemical clock from biogenic NOx emissions around the site. In contrast, we exclude the Manaus plume related data points during the high biomass burning season (IOP2) to ensure that the analysis reflects only the near-field (fresh) and far-field (aged) fires. The results presented here are not strongly sensitive to this data filtering. However, we acknowledge that recirculation in the basin surrounding the T3 site may complicate the quantitative interpretation of the photochemical clock. .Figure 10b demonstrates that under urban influence, there is little evidence for a decrease in BrC absorption due to photochemical processing, similar to our results from Cape Cod. However during the biomass burning season we observe a clear decrease in OA-MAC with increasing aging time (Figure 10c). This suggests that the absorption of OA emitted from biomass burning decreases due to photolysis or oxidation of BrC. We note that the total mass of OA does not increase or decrease significantly with increasing aging time, therefore while we cannot definitively rule out the formation of SOA during transit leading to a "whitening" of total OA, the observations do not suggest that this is a major effect. The binned median OA-MAC values in Figure 10c can be fitted exponentially with $-log(NO_x/NO_y)$ ($R^2$ = 0.95). By assuming the oxidation rate of $NO_2$ to be $1.05 \times 10^{-11}$ $cm^3molec^{-1}s^{-1}$ at 1atm and 300K (Sander et al., 2011) and typical daytime OH concentration of $5 \times 10^5$ molec cm$^{-3}$ we estimate an e-folding time and half-life for the OA absorption to be 22 daytime hours and 14 daytime hours (or 45 hours and 30 hours, assuming zero nighttime OH extending 13 hours of the day during IOP2). Since the biomass burning emissions during IOP2 are frequent and disperse, it is not clear how much of the transport of biomass burning plumes from source to the T3 site occurs at night. These calculated lifetimes neglect other losses of NOx and are highly sensitive to the assumed OH concentration; at extremely high ($\sim 10^7$ molec cm$^{-3}$) or lower ($\sim 10^5$ molec cm$^{-3}$) OH concentrations, the half-life in sunlight would be calculated to be 1 hour or 3.5 days. If the OH concentration in the measurement period is, as assumed here, similar to typical global mean values (Stone et al., 2012), the half-life is comparable to the biomass burning plume observed by Forrister et al., (2015) (9 – 15 hours). However, this half-life would be significantly longer than estimated in laboratory studies which have focussed on SOA aging (5 minutes to 3 hours), suggesting that additional laboratory studies are necessary to examine the aging of BrC from biomass burning. We also observe that the BrC absorption does not appear to decrease to 0 with continued aging (> 15 hours) in our analysis. This is also consistent with Forrister et al., (2015), who suggest that sunlight shows no effect on BrC absorption after about 12 hours of continuous exposure. This may results in a persistent BrC background from fire emissions, even after aging.

## 5. Discussion

By using a new method to derive BrC absorption we identify consistent BrC characteristics from both the global AERONET sunphotometer network and 8 surface sites. At most sites, the BrC absorption contribution in the UV ranges from 10% to

30%. This range of BrC absorption contribution can be used to constrain model simulations or provide a rough estimate of BrC based on measured BC.

The relatively consistent contribution of BrC to total absorption can be explained by the correlation between OC-MAC and the BC/OC mass ratio, which is observed at both AERONET and surface measurement sites. As our analysis shows that higher OC-MAC is found to be associated with lower OC mass contribution in the atmosphere, the BrC absorption contribution lies within a narrow range globally despite differences in the emission. Based on this correlation and BC/OC emission ratios, we estimate a range of MAC: 0.1-3.1 $m^2$/g for OC and 0.05-1.5 $m^2$/g for OA at 440nm from AERONET observations when assuming OA:OC = 2.1. This correlation also suggests the BC/OC emission ratio could provide important information for building an emission inventory for BrC. However our analysis is based on ambient measurements of BC/OC absorption ratios, and extrapolation of these results to emissions require further investigation of how the BC/OC ratio changes (via chemistry, transport, and removal) from source to ambient measurement. Further laboratory studies which include both BrC and BC measurements are required to examine how BrC absorption varies with emission properties.

The BrC-AAE$_{388/440nm}$ that we estimate from AERONET and OMI measurements is also very similar to that which we derive from surface in situ measurements. Both analyses suggest that the global mean BrC-AAE$_{388/440nm}$ is likely to be ~ 4. However, lower AAE$_{388/440nm}$ (<2) are found in Europe from both surface measurements and AERONET sites, suggesting that the BrC in Europe may exhibit different optical properties. These BrC-AAE values are within the range of 1.9 to 9 measured in laboratory experiments (Laskin et al., 2015). However, these results are based on different wavelengths pairs and are therefore not directly comparable. In addition, many of the laboratory reported values typically rely on one long wavelength in the visible spectrum (e.g. AAE$_{370/660nm}$ or AAE$_{330/600nm}$). Since BrC absorption at these long wavelengths is too small to be detected accurately, these BrC-AAE are likely quite uncertain.

For well-mixed air masses exposed to urban emissions, our analysis of the observations at Cape Cod in February and at GoAmazon-T3 site in the non-biomass burning season do not provide any evidence for evolving BrC optical properties associated with photolysis or oxidation. It may be that either the chemistry impact is not as significant as emissions/transport or that the photolysis and generation of BrC counteract each other. In contrast, the observations at GoAmazon-T3 site during the biomass burning season exhibit a ~1 day photochemical lifetime (in sunlight) for BrC absorption. This decrease in absorption is qualitatively consistent with previous field observations (Forrister et al., 2015) and may suggest that the absorption of BrC from fire emissions is geographically limited to the near-field. This may also explain the somewhat counter-intuitive lack of strong BrC signature in biomass burning regions/seasons in our analysis of AERONET observations (Figure 3). The majority of studies which have investigated the "browning" or "whitening" of BrC have focused on laboratory experiments at extreme conditions, however, the chemical processes in the real atmosphere may be very different from these controlled environment. Additional laboratory and field studies of how the optical properties of primary BrC may evolve due to photoxidation are required.

Using a BrC-AAE = 4 as suggested by our analysis of AERONET and OMI satellite observations, the BrC column absorption contribution is 0-40% at 440nm and less than 20% at 550nm. This suggests that the previous model estimated

BrC absorption DRE contributions (20% - 40%) are likely to be biased slightly high (Feng et al., 2013; Lin et al., 2014). Including photochemical "whitening" of BrC from fires in these models may resolve these discrepancies.

By applying a new AAE method that we describe in this paper, we have obtained global observational constraints on BrC absorption. However, these results are subject to uncertainties associated both with the methodology and with the dataset to
which it is applied. The core issue for all methods that use the AAE to estimate the absorption from BrC is the uncertainty associated with the absorption of BC. Our method improves upon previous studies using this approach by using the information from the wavelength dependent measurements themselves and by allowing for an atmospherically-relevant range of BC properties, rather than fixing these at a single assumed value. Additional constraints on BC optical properties and mixing state would help further improve the method.
Given the large uncertainties associated with AERONET retrievals of AAOD, the most challenging aspect of our analysis is that an accurate globally distributed multiple-wavelengths aerosol absorption measurement data set is unavailable at present. Thus, while our study provides qualitative global constraints, and insight into BrC aging processes from the aethalometer observations, achieving a better understanding of the properties, evolution, and impacts of global BrC will rely on the future deployment of accurate multiple-wavelength absorption measurements to which AAE-methods, such as the approach
developed here, can be applied.

**Acknowledgements.** This work was supported by the EPA-STAR program. Although the research described in this article has been funded in part by the US EPA through grant/cooperative agreement (RD-83503301), it has not been subjected to the Agency's required peer and policy review and therefore does not necessarily reflect the views of the Agency and no
official endorsement should be inferred. The GoAmazon2014/5 and TCAP data were obtained from the Atmospheric Radiation Measurement (ARM) Climate Research Facility, a U.S. Department of Energy Office of Science user facility sponsored by the Office of Biological and Environmental Research. For the GoAmazon2014/5 data, we also acknowledge the support from the Central Office of the Large Scale Biosphere Atmosphere Experiment in Amazonia (LBA), the Instituto Nacional de Pesquisas da Amazonia (INPA), and the Universidade do Estado do Amazonia (UEA). The work of
GoAmazon2014/5 campaign was conducted under 001030/2012-4 of the Brazilian National Council for Scientific and Technological Development (CNPq). The AMS measurement at GoAmazon2014/5-T3 site was performed using EMSL, a DOE Office of Science User Facility sponsored by the Office of Biological and Environmental Research and located at Pacific Northwest National Laboratory. We thank Jesse Kroll, Eleanor Browne, Kelsey Boulanger, Anthony Carasquillo, and Kelly Daumit for AMS measurement during TCAP. We also thank the Norwegian Institute for Air Research (NILU) and the
NOAA Earth System Research Laboratory for providing the EMEP and NOAA/ESRL background site measurements, and the AERONET staff for establishing and maintaining the sunphotometer network used in this study.

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

**Table 1: Summary of surface sites measuring aerosol absorption by aethalometer**

| Site | Location (latitude, longtitude) | Time period | Site property | Mean aerosol absorption at 370nm, Mm$^{-1}$ | Campaign | References |
|---|---|---|---|---|---|---|
| Zeppelin Mountain | (78.9N, 11.9E) | 2010 | background | 0.63 | EMEP[a] | http://www.emep.int |
| Barrow | (71.3N, 156.6W) | 2010 - 2014 | background | 1.31 | NOAA/ESRL site | http://www.arm.gov |
| Tiksi | (71.6N, 128.9E) | 2012 - 2014 | urban | 3.8 | NOAA/ESRL site | http://www.arm.gov |
| Cool | (38.9N, 121W) | Jul.2010 | rural | 7.5 | CARES[b] | Zaveri et al., 2012 |
| Finokalia | (35.3N, 25.7E) | 2014 | background | 13.54 | EMEP | http://www.emep.int |
| Ispra | (45.8N, 8.6E) | 2010 - 2011 | urban | 121.6 | EMEP | http://www.emep.int |
| Prelia | (51.4N, 21E) | 2010 | urban | 29.29 | EMEP | http://www.emep.int |
| SIRTA | (48.7N, 2.16E) | Sep.2010 – Mar.2011 | urban | 55.95 | EMEP | http://www.emep.int |
| GoAmazon-T3[c] | (3.2S, 60.6W) | 2014 - 2015 | urban/rural | 5.59 | GoAmazon2014/5 | Martin et al., 2015 |
| Cape Cod | (42.3N, 70.1W) | Jul.2012; Feb.-Mar.2013 | urban/background | 9.22 | TCAP[d] | Berg et al., 2016 |

[a] EMEP: The European Monitoring and Evaluation Program
[b] CARES: Carbonaceous Aerosols and Radiative Effects Study
[c] T3 site in GoAmazon2014/5 (Observations and Modeling of the Green Ocean Amazon ) campaign
[d] TCAP: Two-Column Aerosol Project

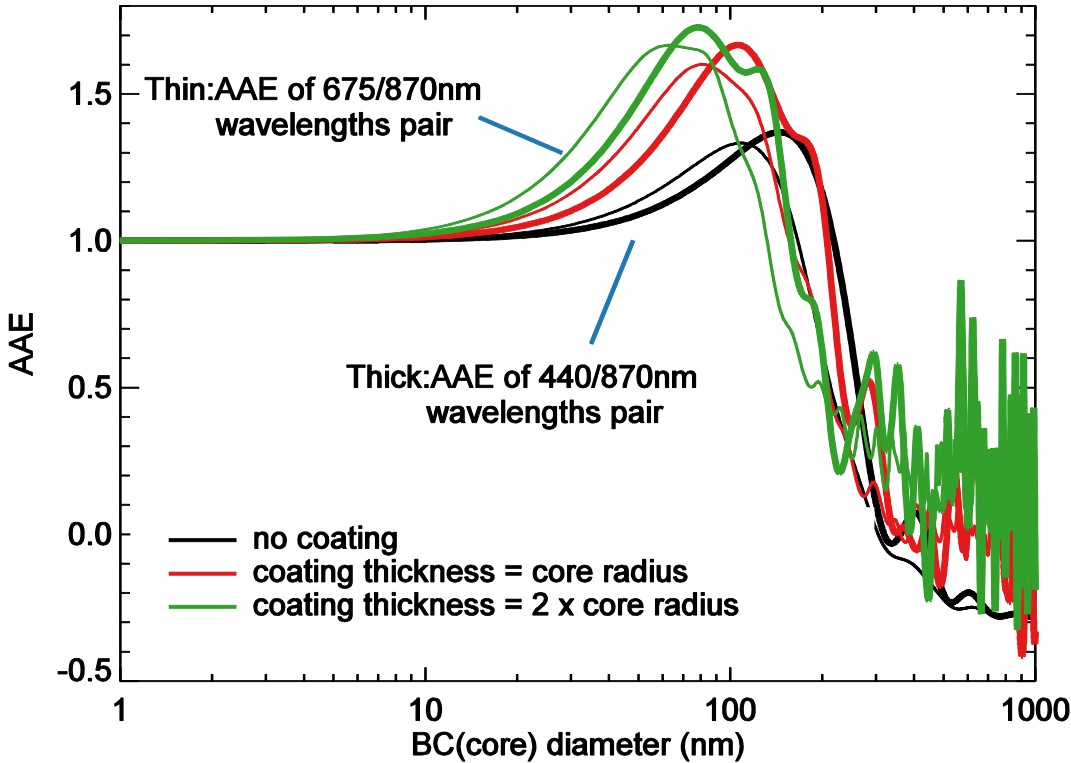

**Figure 1: The Angstrom absorption exponent (AAE) for BC estimated using Mie calculations as a function of size and for a series of coating states.**

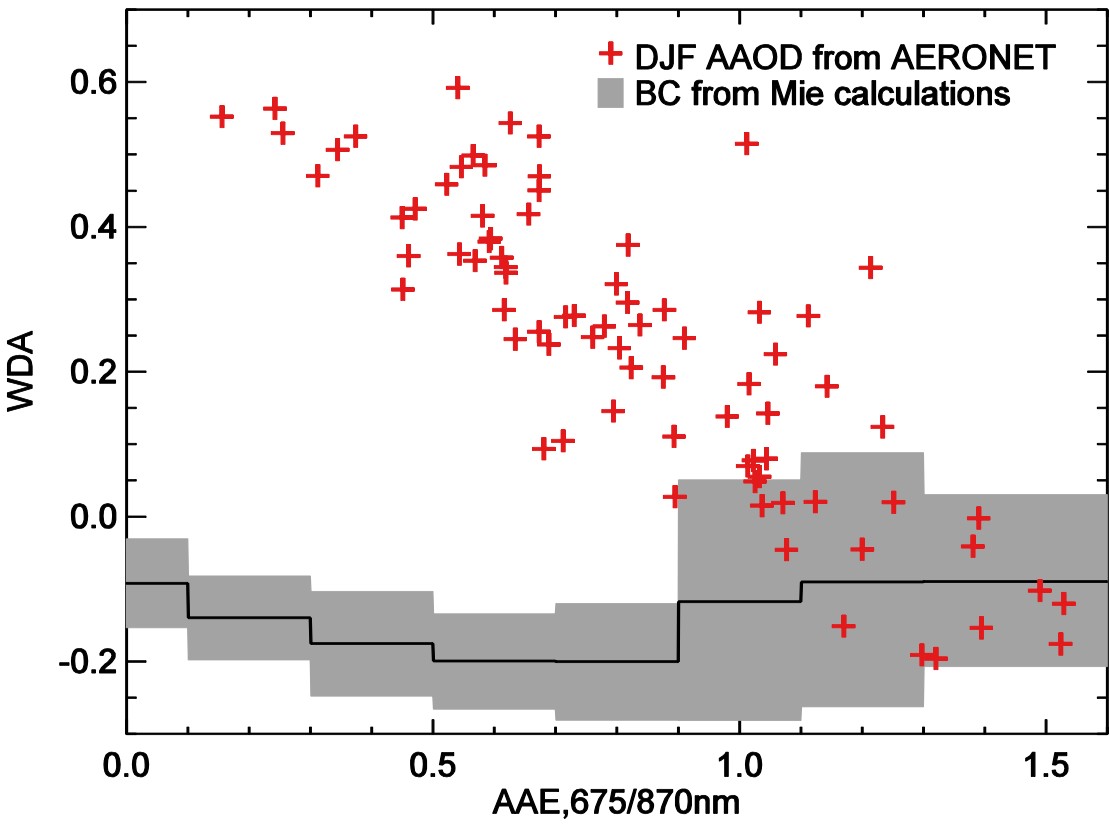

**Figure 2: The range in the estimated wavelength dependence of AAE (WDA) for BC (shaded region) based on Mie calculations (see Section 2 for size and coating assumptions). The black line is the median WDA. Red crosses show the total absorption from 2005-2014 10 years seasonal average AAOD measurements at 3 wavelengths from the AERONET network in north hemisphere winter (December, January and February). Observations which lie above the shaded region include detectable contributions of BrC absorption.**

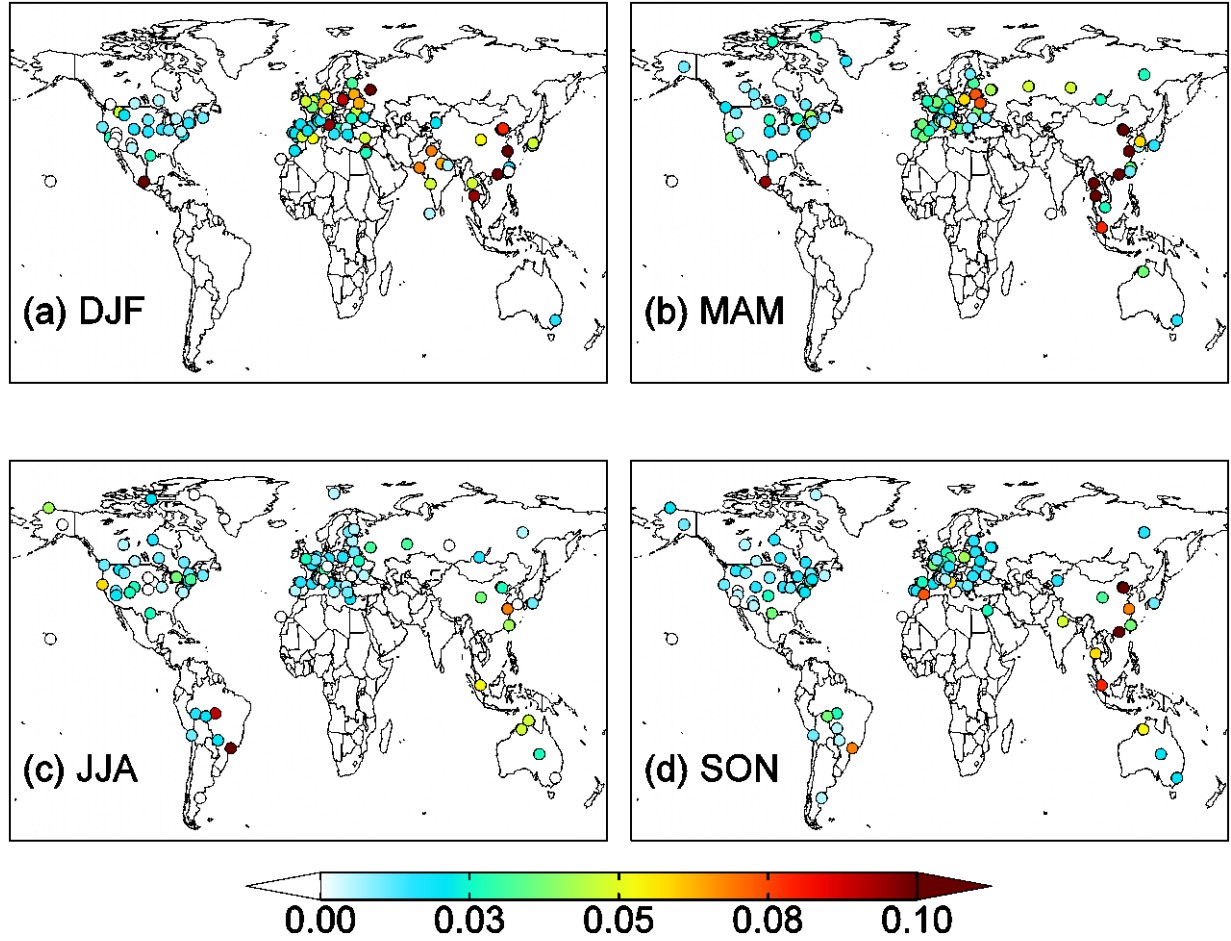

**Figure 3: Derived seasonal mean BrC-AAOD at 440nm from AERONET observations (2005-2014, 10 years average) in northern hemispheric (a) winter, (b) spring, (c) summer, and (d) fall. The color bar is saturated at 0.010 to emphasize regional variations, but maximum values reach 0.056.**

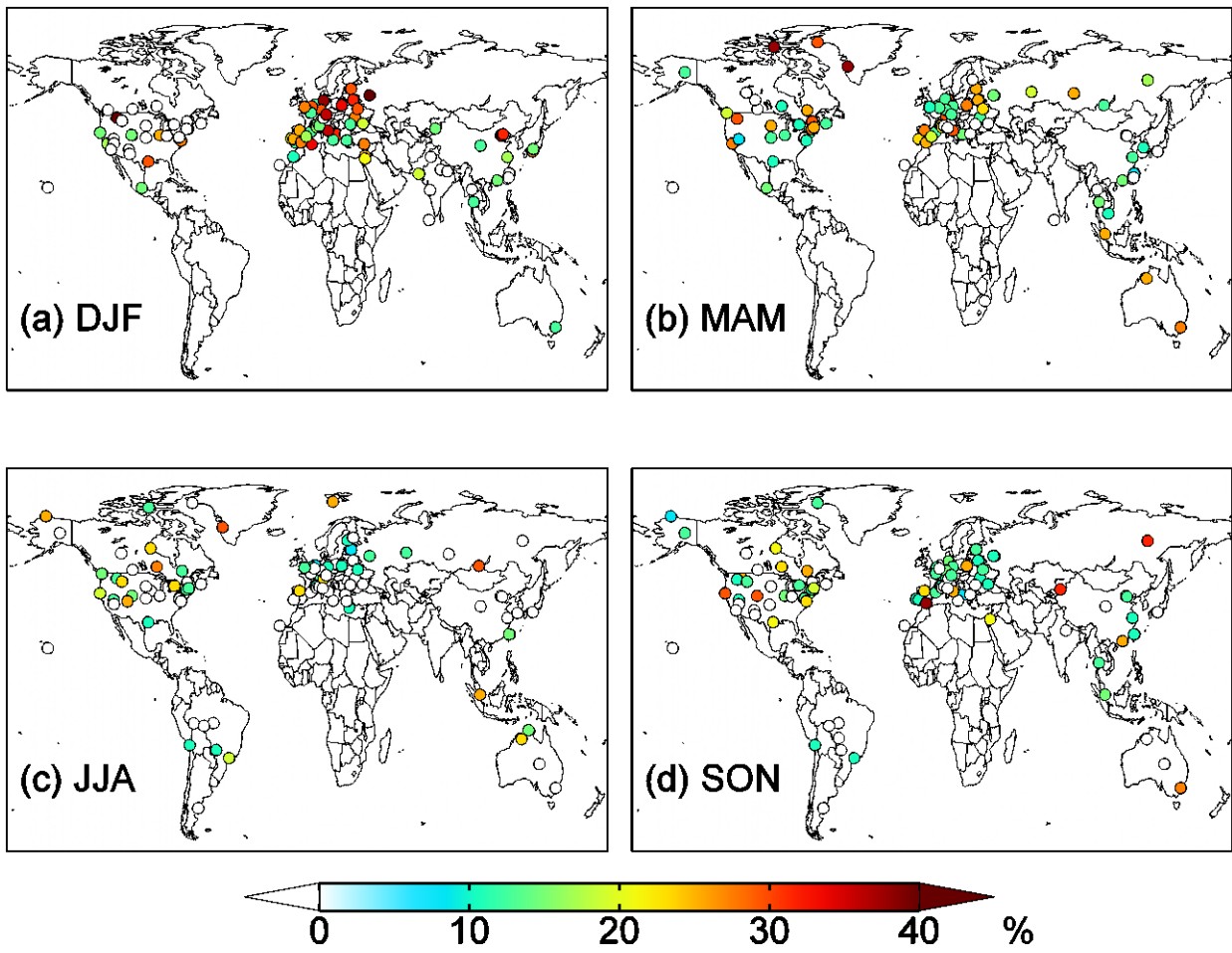

**Figure 4: Derived seasonal mean BrC-AAOD contributions to total AAOD at 440nm from AERONET observations (2005 -2014, 10 years average) in northern hemispheric (a) winter, (b) spring, (c) summer, and (d) fall. The color bar is saturated at 40% to capture regional variation, but values up to 45% are estimated at specific sites.**

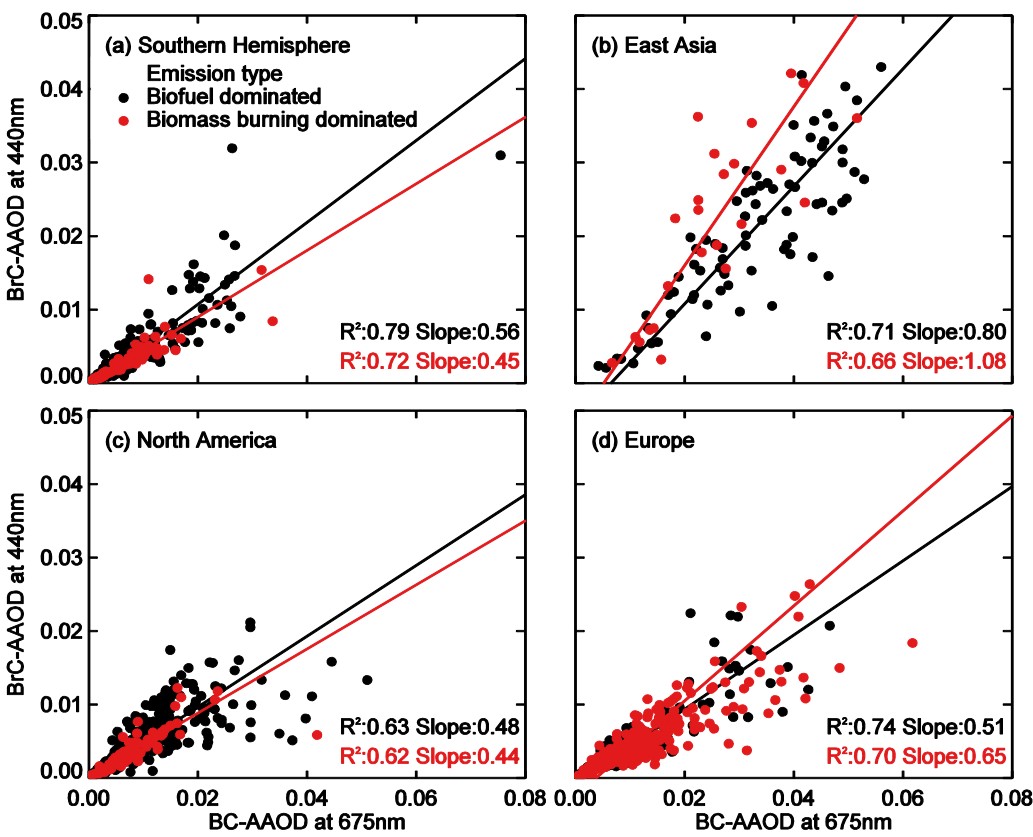

**Figure 5:** The relationship between monthly mean derived AERONET BrC-AAOD at 440nm and BC-AAOD at 675nm at AERONET stations in (a) North America, (b) East Asia, (c) Europe, and (d) Southern Hemisphere for the years 2005-2014.

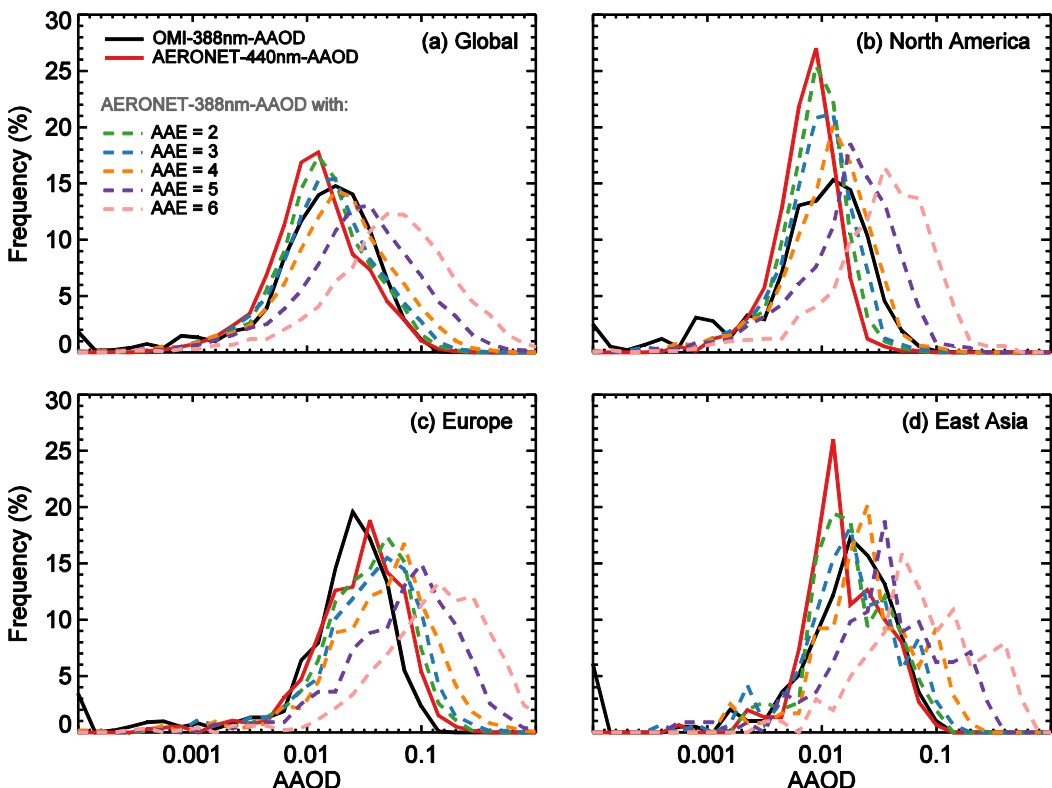

**Figure 6: Frequency distributions of the AAOD measured by OMI at 388 nm (black) and AERONET at 440 nm (red), as well as AERONET observations adjusted to 388nm using a range of assumed AAE for BrC (dashed lines) for (a) the whole world, (b) North America, (c) Europe and (d) East Asia. Details can be found in Section 3.4.**

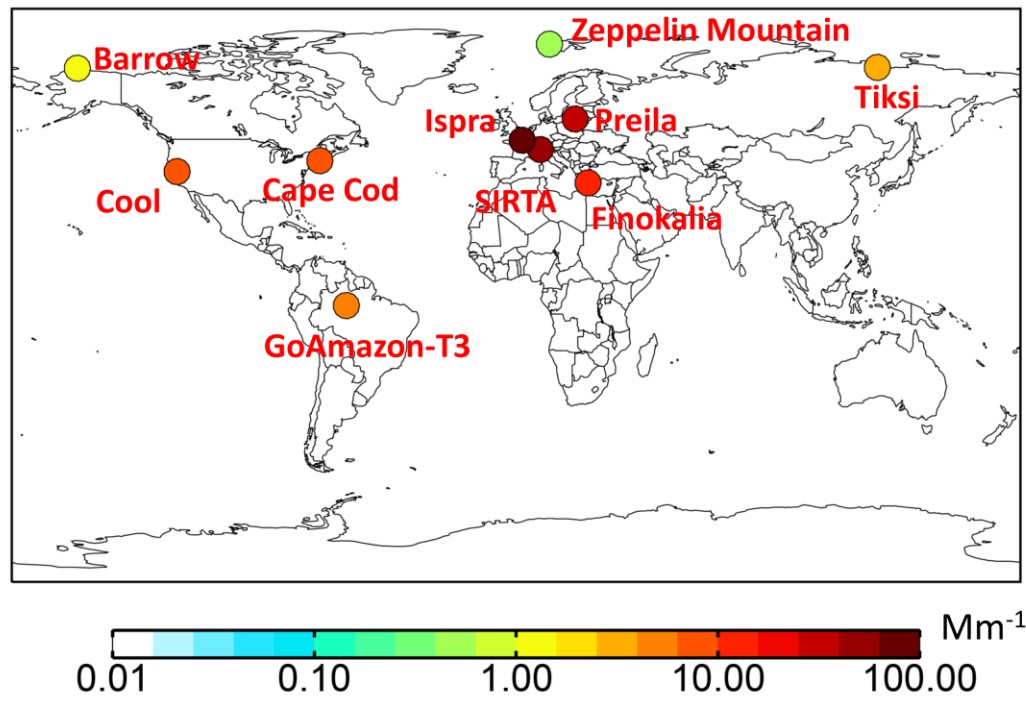

**Figure 7: The locations of a series of field sites used in this study where multiple-wavelengths absorption was measured using an aethalometer. Each site is colored by the mean aerosol absorption measured at 370nm. Values from the GoAmazon-T3 site are corrected, while others are uncorrected (see Section 4.1).**

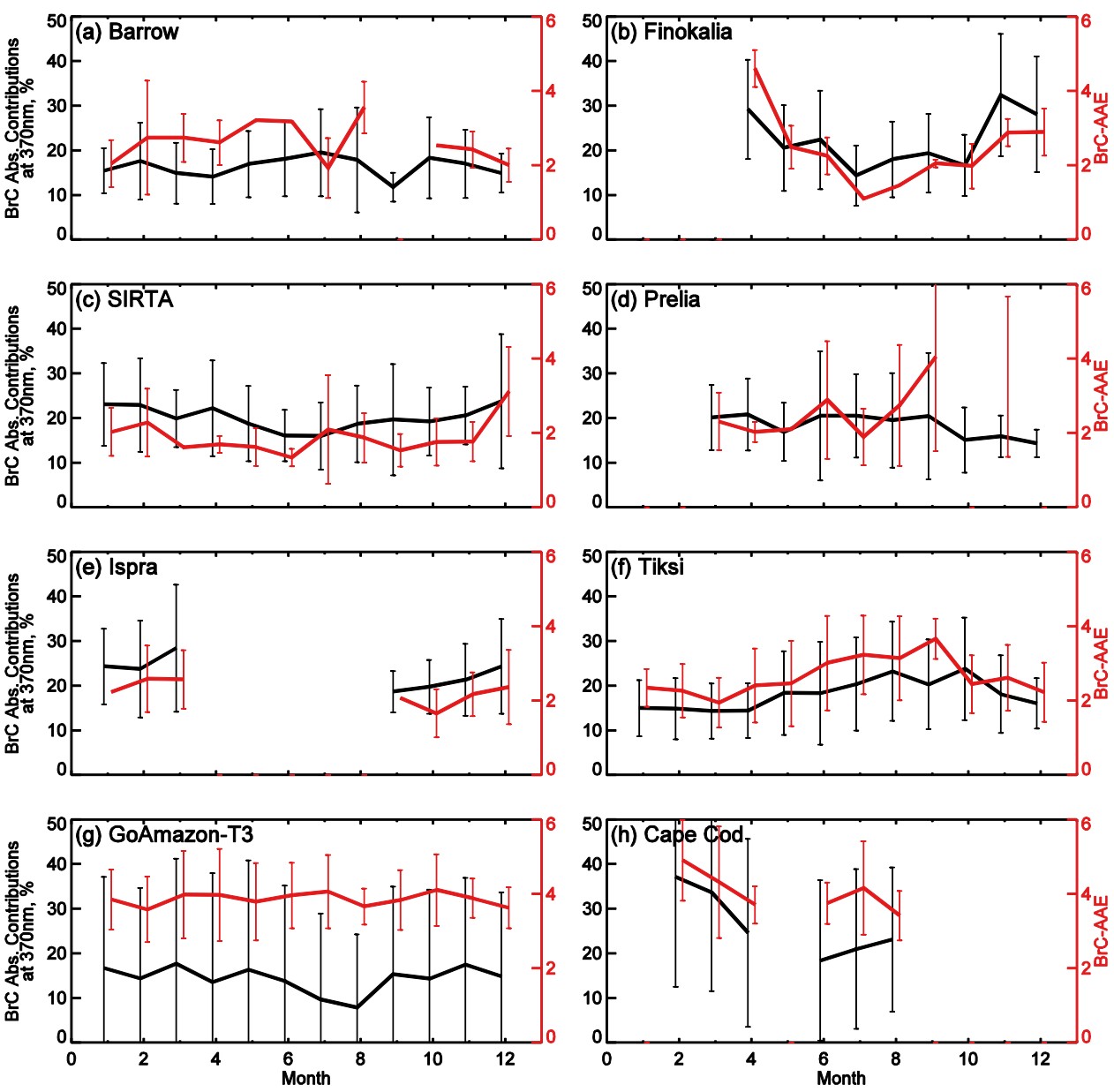

Figure 8: Monthly variation in the derived BrC contribution to total absorption at 370nm (black) and BrC-AAE (red) at a series of surface sites (see Table 1, Figure 7). The BrC-AAE in (g) GoAmazon-T3 site and (h) Cape Cod are calculated using the 370-430nm wavelength pair, those at other sites are based on the 370-470nm wavelength pair. Error bars indicate the standard deviation for values averaged in each month.

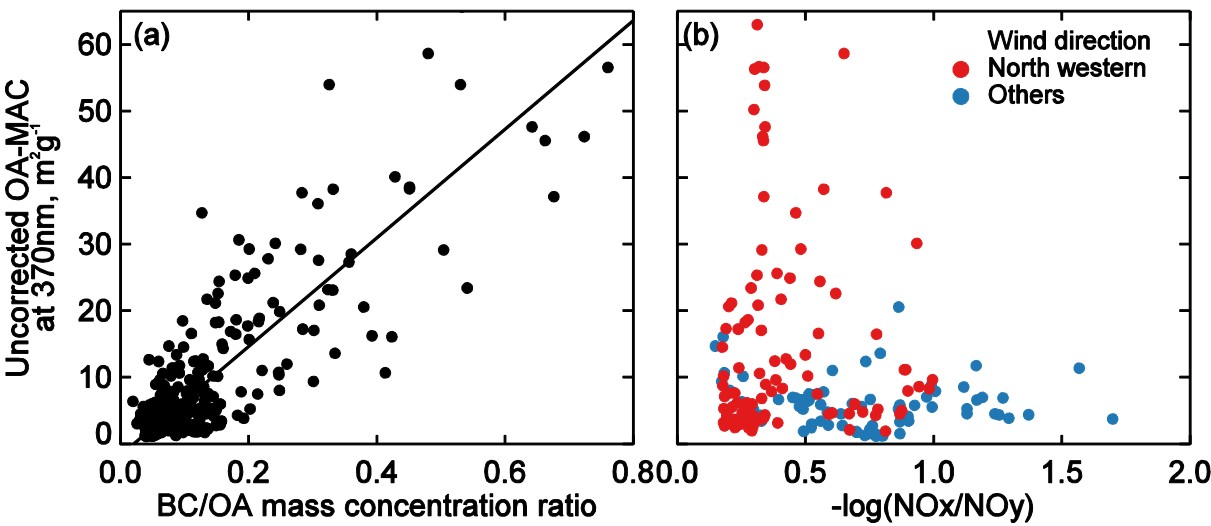

**Figure 9: Measurements from the TCAP campaign in Cape Cod in February 2013. The relationship between uncorrected OA-MAC measured at 370nm and (a) the observed BC/OA ratio, (b) measured photochemical clock. All values are hourly means. The colours in (b) indicate different wind directions discussed in Section 4.3.**

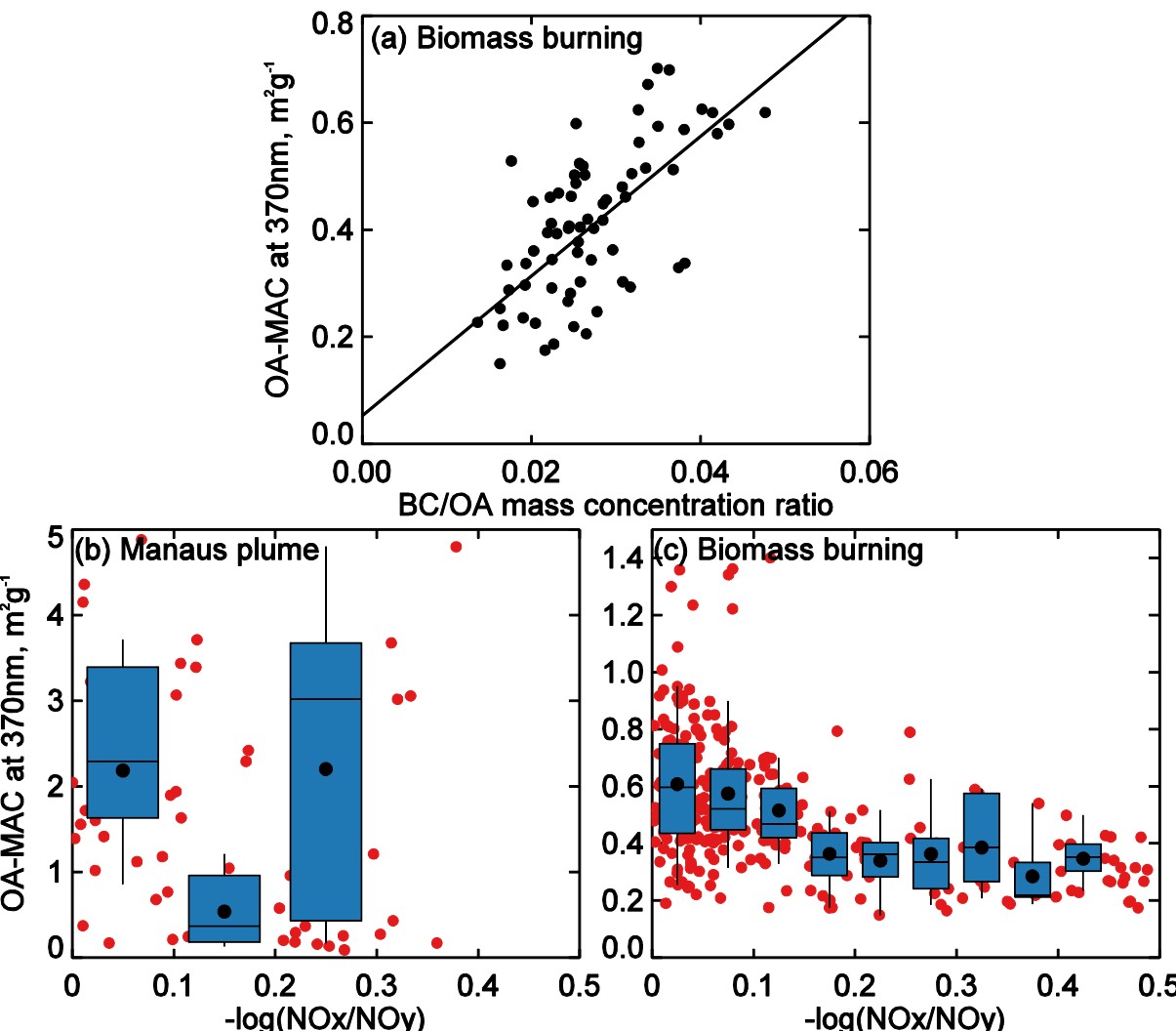

**Figure 10: Measurements from the GoAmazon2014/5 campaign at the T3 site, to the west of Manaus, Brazil. The relationship between hourly mean BrC-MAC measured at 370nm and (a) the observed BC/OA ratio during August 15[th] to October 15[th] in 2014 (IOP2), (b) measured photochemical clock during January to March in 2015 with selected Manaus plumes and (c) IOP2, excluding Manaus plumes. In (b) and (c), the red points are individual hourly-average measurements, whereas the box and whisker plots show the binned mean (black points), median (black lines inside boxes), lower and upper quartile (boxes), and 5[th] and 95[th] percentile (whisker). The number of data points (N) are shown inset.**