# Peer review of "Deriving Brown Carbon from Multi-Wavelength Absorption Measurements: Method and Application to AERONET and Aethalometer Observations"

_Atmospheric Chemistry and Physics, 2016_

## Referee Comment (RC1) · Anonymous Referee #1 · 14 Apr 2016

Review of "Deriving Brown Carbon from Multi-Wavelength Absorption Measurements: Method and Application to AERONET and Surface Observations" by Xuan Wang, Colette L. Heald, Arthur J. Sedlacek, Suzane S. de Sa, Scot T. Martin, M. Lizabeth Alexander, Thomas B. Watson, Allison C. Aiken, Stephen R. Springston and Paulo Artaxo

This paper uses empirical methods to derive organic carbon absorption from optical and in situ measurements. The authors apply their results to hundreds of AERONET sites and 10 field sites. Unfortunately, their methodology is not sound, and their results are not consistent with what is known about BrC.

MAIN POINTS

The authors are merely adding a new twist to the AAE method pioneered by Chung (2012) and Bahadur (2012) (although they did not acknowledge the similarity of their work to these previous works). The authors determine the AAE in two spectral regions via Eq 1, and then define the nonlinearity of the AAE as the wavelength dependence of the AAE (WDA) in their Eq 2. Their WDA is presented as a ratio of two exponentials, so mathematically it is equivalent to

$$WDA = \exp(AAE\_440/870 - AAE\_675\text{-}870).$$

Later, the authors use ln(WDA) in their Eq 4, so they are really just using the difference of AAE measured over two spectral regions to characterize the BC and BrC absorption at 440 nm. This is very similar to Bahadur (2012)!

The authors go on to compute WDA for black carbon (BC) using a set of lognormal distributions (p4, lines 22-31), and provide a range of WDA values for their computations as the grey shaded area in their Figure 2. The median of this range of values is the black line, which represents the $WDA\_0$ of Eq 4. Thus, the authors are using theoretical BC AAE calculations as a baseline AAE; differences from this baseline are attributed to BrC. Particle size effects on AAE are not considered, although extreme dust cases are later omitted from the analysis (EAE < 1 and AAE > 1.5).

There are many problems with this. First of all, the description of the parameters used for their Mie calculations does not include enough details for a reader to understand and duplicate the shaded region of figure 2. For instance, how thick are the coatings when the absorption enhancement factor is 2? What are the optical properties of the coatings? How does varying the optical properties of the coatings affect the size of the shaded area? Which size distributions represent the baseline? Are these 20nm particles, or 300 nm particles? Do the baseline particles have coatings? How thick? Which AERONET sites are falling inside of the shaded region of Fig 2? Urban OC is non-absorbing, so the urban aerosols should fall inside of the shaded region, and the biomass burning aerosols should fall outside of the shaded regions. Looking at Figs

3&4, I don't believe that this is the case.

How come the shaded area of Figure 2 does not include the point (1,1), which would represent AAE = 1 for the entire wavelength range? Some of the particles sizes for their calculations are small enough (20 nm GMD) to reach the theoretical limit for small absorbing particles.

Why aren't any coarse mode aerosols included in the computations of the shaded region? There is always a coarse mode in the AERONET retrievals, and this will affect AAE. Sure, the authors eliminate aerosols dominated by dust in Section 3.1 by filtering for EAE < 1, but aerosol systems with 1 < EAE < 2 are mixtures of fine and coarse particles. Thus, there is dust in much (most?) of their analysis.

How well is the AERONET AAOD known? How does the uncertainty in AAOD at each wavelength affect the uncertainty in the AAE in the two spectral regions that you are using? How do the AAE uncertainties affect the BrC AAOD retrieval?

Beyond the theoretical issues with this approach, though, the results do not seem to back up this approach. Figure 3 shows regional maps of seasonal mean BrC AAOD. The biomass burning sites in S. America never show elevated BrC AAOD, and the biomass burning seasons are not detectable over Africa.The Asian sites seem to have elevated BrC in all seasons except for Summer.

Figure 4 provides seasonal mean fractions of (BrC AAOD) / (total AAOD), and those results are even more ambiguous. The colors on these maps are all very random, with dark reds co-mingled with light blues; one would expect neighboring AERONET sites to exhibit similar colors if they were representing similar source regions. The randomness of the colors on these maps, though, seem to indicate noise.

OTHER IMPORTANT POINTS:

Page 3, line 6: AERONET provides *retrievals*, not measurements.

Page 3, line 23-25: Chung (2012) and Bahadur (2012) do not assume AAE = 1 for

BC, as stated here. The authors need to be more careful when attributing concepts to others.

Page 5, line 8: Bahadur (2012) has nothing to do with the SSA screening conditions of AERONET. Cite Dubovik (JGR, 2000), instead.

PAge 5, line 16: The authors state: "... and cannot be estimated using our method without additional information about the size and coating state of the particles." However, you have information about size in the AERONET retrievals. Why not use them?

Page 5, equations 3&4: These equations should be reversed. WDA_0 is used to compute BC AAE_440/870, which is used to compute BC abs(440), right? So presently, Eq 4 gets used before Eq3.

Page 6, line 18: The authors state: "In general, BrC-AAOD is smaller than 0.005 at most sites but larger in Asia and occasionally in northern Africa and South America." BrC-AAOD < 0.005 in general? That sounds like a small number. Does the AERONET retrieval really have the sensitivity to quantify this parameter, then? AERONET only claims an accuracy of ∼0.01 for AOD; the accuracy of AAOD is certainly no better. So all of the BrC-AAODs that you are retrieving seem to be well below AERONET's accuracy threshold.

Page 6, lines 24-26: The authors are quoting numbers for seasonal variability, but they do not specify seasons or sites. The colors on the maps do not obviously correspond to the numbers quoted here. The authors should at least give these "special" sites a different shape, so that the reader can understand which sites are the "African biomass burning sites." Also, the biomass burning season on the Sahel is not July-October. It starts in November and ends in the Spring.

Page 6, line 31: The authors state: "In winter the BrC AAOD contributions in North America (15%), Europe (17%) and East Asia (23%) are larger than in summer (10%, 13% and 14%)." Why should BrC AAOD fractions be higher in the winter in North America? Few Americans use biofuel, and the biomass burning season is in the summer. Also, higher winter BrC AAOD fractions are not obvious from Fig 4.

Equation 6: The authors assume that all OC are BrC for Equation 6. This is not a valid assumption.

PAge 12, line 10: The authors state "the correlations between BrC-AAE and BrC absorption contributions are only slight to moderate at these sites." They should still provide the correlations to the readers, though.

PAge 12, lines 22-23. The authors state that OA-MAC is correlated with BC/OC mass ratio, but they do not help the reader to understand why this should be the case (anywhere in the article). Thus, for a given OC mass, the OA-MAC is linearly related to BC mass? This sounds like BC is contaminating their OC-MAC retrieval (rather than proof that their retrieval is working).

PAge 13, lines 26-28 and Figs 10b, 10c: Authors find that the OA-MAC for the non-biomass plumes (Fig 10b) are much much higher than for the BB plumes (Fig 10c). WHY? OA-MAC for urban aerosols should be negligible.

---

## Referee Comment (RC2) · Anonymous Referee #2 · 26 Apr 2016

*This paper attempts to quantify brown carbon absorption using the AERONET data base of aerosol extinction optical depth and single scattering albedo as well as aethalometer measurements at a number of stations. The approach used in this analysis has been used in previous studies [Schuster et al., 2005; Russell et al., 2010, to mention a few], strangely ignored in the literature review. An innovative component of this work is the combined use of surface and space based AAOD retrievals for the estimation of aerosol absorption exponent. There are several issues in the paper that need to be addressed.*

*The use of annual average AAOD values in Figure 2, ignores the seasonal variability of aerosol type. It would be more illustrative to plot instead seasonally averaged data and identify in a table the sites used in the analysis. I suspect the analysis presented in section 2 will change substantially if seasonal AAOD data is used.*

*The seasonal maps showing the 10-year average of seasonal mean BrC-AAOD (as indicated in the caption of Figure 3) look very strange to me. I wonder if that is really what is being plotted. I am very surprised by the number of AERONET sites having 10 years of observations over the 2005-2014 period. Most sites shown on those maps do not have a continuous ten year record. Surprisingly the seasonal maps of derived BrC-AAOD based on AERONET observations do not show the expected AAOD hot spots in the Sahel (DJF), Southern Africa (JJA, SON), and Amazon Basin (SON). The only known hot-spot correctly showing is the Southeast Asia (MAM). I suspect these maps are more a representation of the geographical and temporal bias of AERONET observations over the 2005-2014 period than an actual representation of the AAOD load as intended by the authors. The number of sites in SE Asia and China has increased significantly over the last ten years. On the other hand, the coverage over Southern Africa has gone down significantly. A better description and explanation of this map is needed. I suggest to re-do these maps using a more careful selection of sites making sure that the temporal coverage is similar for all used stations.*

*Other comments:*

*Page 6, line 24. Abundant biomass burning also takes place in Northern South-America and the Sahel during NH winter. Also in the NH Spring there is intense biomass burning in Southeast Asia and Central America.*

*Page 9, line 1. Add the Torres et al [2014] reference that describes the latest algorithm upgrades.*

*Page 9, line 8. Nowhere in the quoted references, is it said that OMI and AERONET AAOD have a correlation > 0.8 as stated by the authors. They incorrectly imply that OMI and AERONET AAOD have been directly compared citing papers that refer to AOD validation. Both OMI AOD [Ahn et al. 2014] and SSA [Jethva et al, 2014] have been compared to AERONET produced values. The authors need to read the papers and correctly quote them.*

*The title of the paper is confusing because both AERONET and aethalometer data are obtained by means of surface observations. The title should also reflect the fact that OMI observations are used in the estimation of the reported results. Suggest changing title to 'Deriving brown carbon absorption from multi-wavelength absorption measurements: Method and applications to AERONET, OMI and Aethalometer observations'*

*References:*

*Russell, P. B., Bergstrom, R. W., Shinozuka, Y., Clarke, A. D., DeCarlo, P. F., Jimenez, J. L., Livingston, J. M., Redemann, J., Dubovik, O., and Strawa, A.: Absorption Angstrom Exponent in AERONET and related data as an indicator of aerosol composition, Atmos. Chem. Phys., 10, 1155-1169, doi:10.5194/acp-10-1155-2010, 2010.*

*Schuster, G. L., O. Dubovik, B. N. Holben, and E. E. Clothiaux (2005), Inferring black carbon content and specific absorption from Aerosol Robotic Network (AERONET) aerosol retrievals, J. Geophys. Res., 110,* D10S17, doi:*10.1029/2004JD004548*.

*Torres, O., Ahn, C., and Chen, Z.: Improvements to the OMI near UV aerosol algorithm using A-train CALIOP and AIRS observations, Atmos. Meas. Tech., 6, 5621-5652, doi:10.5194/amtd-6-5621-2013, 2013.*

*Jethva, H., O. Torres, and C. Ahn (2014), Global assessment of OMI aerosol single-scattering albedo using ground-based AERONET inversion, J. Geophys. Res. Atmos., 119, doi:10.1002/2014JD021672.*

---

## Referee Comment (RC3) · Anonymous Referee #3 · 28 Apr 2016

This work provides an observational constraint for brown carbon aerosols (BrC), which absorb solar radiation, and thus have an important implication for climate. Based on the previous study for quantifying the brown carbon aerosol absorption, authors used observed absorption angstrom exponents at a pair of wavelengths at AERONET sites to reduce associated uncertainties. I believe that this dataset adds up to the observations of BrC absorptions, which are very sparse globally and thus it will be valuable to evaluate the estimated contributions of BrC to aerosol absorption and radiative forcing. However, the observed quantity derived from the combination of various observations needs additional clarification and the details are listed as follows.

P4, L23 - Authors need to estimate the associated uncertainty with the assumption

[Figure]

of spherical BC in their method. For example, Kahnert and Devasthale et al. (2011) estimated the difference of SSA up to 0.05 between spherical vs. aggregate shapes of BC (Figure 5 in their paper).

P5, L21-24 - Please elaborate how you obtain 4% uncertainty.

Figure 2 - It would be recommended to remove the dust contribution as shown in Section 3.

P7, L14-15 - Several papers showed that BrC absorption at 675 nm is significant (Alexander et al., 2008; Chung et al., 2012). So I am wondering if you assume an absorption at 879 nm as BC absorption alone, then how would your results differ. Or at least, you many need to discuss those previous papers and the possible effect on your estimates.

P7, L19 - Why did you use GFED3 for 2011 and earlier? If there is no reason for this, you better use GFED4 for the entire period consistently.

P9, L8-9 - Jethva and Torres (2011) and Ahn et al. (2014) conducted an evaluation of AOD alone, not AAOD. Jethva et al. (2014) did an SSA evaluation and showed that OMI SSA are higher than AERONET SSA. For example, about 50% of total samples showed the difference of 0.03 or higher and 25% showed 0.05 or higher differences. This is a considerable discrepancy between two datasets and may have a huge impact on your estimates. For example, if AERONET SSA is 0.94 and OMI is 0.97, then the estimated AAODs using AERONET versus OMI data differ by 100%. So BrC AAE using OMI and AERONET AAOD together may cause too high uncertainty. Please consider a bias correction for OMI or simply drop out OMI data in your calculation.

minor corrections,

P2, L28: Forrister et al., (2015) -> Forrister et al. (2015) P7, L15: 675 m -> 675 nm
* * *

---

## Short Comment (SC1) · 24 May 2016

I would like to correct a statement made by Reviewer#3 about the OMI SSA evaluation using AERONET inversion dataset as reported in Jethva et al. [2014].

Reviewer states that "Jethva and Torres (2011) and Ahn et al. (2014) conducted an evaluation of AOD alone, not AAOD. Jethva et al. (2014) did an SSA evaluation and showed that OMI SSAs are higher than AERONET SSA. For example, about 50% of total samples showed the difference of 0.03 or higher and 25% showed 0.05 or higher differences."

A careful examination of OMI versus AERONET SSA plots shown in Jethva et al. [2014]

[Figure]

, page 14, figure 9, suggests that the percentage matchups that falls within $\pm 0.03$ and $\pm 0.05$ uncertainty limits are with reference to their absolute difference (OMI minus AERONET) which includes both, positive and negative biases. This is clearly evident in the scatter-plot in which the matchups are spread on both upper and lower sides of the one-to-one line. Furthermore, in the same Figure 9, the difference between OMI and AERONET SSAs is shown as a function of UV-AI, which further illustrates that though OMI SSAs are overall bias high at lower end of UV-AI range (<2.0), the differences are evenly spaced on both positive and negative sides for UV-AI greater than 2.0. However, the present reviewer misinterprets that OMI SSAs are always bias high with reference to the AERONET SSAs, which seems to be not true.

For the reference, I am including Figure 9 of Jethva et al. [2014] in this comment.

Also, note that the comparison between AERONET and OMAERUV SSA retrievals does not constitute a validation analysis since both measuring techniques are based on inversion algorithms that rely on assumptions. The resulting level of agreement can only be interpreted as a measure of consistency (or lack thereof) in the measurement of the same physical parameter by fundamentally different remote sensing approaches.

Best,

Dr. Hiren Jethva Research Scientist, Universities Space Research Association/NASA Goddard Greenbelt, MD 20771 USA Email: hiren.t.jethva@nasa.gov

[Figure]

[Figure]

**Fig. 1.**

---

## Author Comment (AC1) · 6 Jun 2016

We thank all the reviewers for their time and comments. We have made efforts to improve the manuscript accordingly, please find response for corresponding points below.

**Reviewer #1**

This paper uses empirical methods to derive organic carbon absorption from optical and in situ measurements. The authors apply their results to hundreds of AERONET sites and 10 field sites. Unfortunately, their methodology is not sound, and their results are not consistent with what is known about BrC.

**MAIN POINTS**

The authors are merely adding a new twist to the AAE method pioneered by Chung (2012) and Bahadur (2012) (although they did not acknowledge the similarity of their work to these previous works). The authors determine the AAE in two spectral regions via Eq 1, and then define the nonlinearity of the AAE as the wavelength dependence of the AAE (WDA) in their Eq 2. Their WDA is presented as a ratio of two exponentials, so mathematically it is equivalent to

WDA = exp(AAE\_440/870 - AAE\_675-870).

**Later, the authors use ln(WDA) in their Eq 4, so they are really just using the difference of AAE measured over two spectral regions to characterize the BC and BrC absorption at 440 nm. This is very similar to Bahadur (2012)!**

We respectfully disagree with the reviewer about the similarity of our method with Chung et al. (2012) and Bahadur et al. (2012). We agree that both Chung et al. (2012), Bahadur et al. (2012) and our work focus on the difference of AAE over different spectral wavelengths, however, they are different in many aspects:

1) In Bahadur et al. (2012) (similar for Chung et al., 2012), they describe the total AAOD of carbonaceous aerosols at wavelengths  $\lambda$  as below:

$$AAOD(\lambda) = AAOD_{ref,BC}(\lambda/\lambda_{ref}) - ^{AAEBC} + AAOD_{ref,OC}(\lambda/\lambda_{ref}) - ^{AAEOC}$$
(1)

They then used observations to find the empirical AAEBC and AAEOC to solve the equation.

In contrast, we do not use the AAE of either BC or OC in our approach. Given the poor understanding of BrC, we do not think it appropriate to assume the properties of BrC (e.g. AAEOC) in a BrC derivation method. However, BC optical properties are better understood, which makes it possible to find a range for BC-WDA. Then the possible range of BC absorption (as well as BrC absorption) can be estimated using the relationship in Figure 2.

The core idea of our method is to find the range of BC absorption in any multiple-wavelengths absorption measurements. We do not try to characterize BrC optical properties in our method. The most important result of our method is the theoretical relationship of BC-WDA and BC-AAE shown in Figure 2, which to our best knowledge, has not been discussed in any previous studies (including Chung et al., 2012 or Bahadur et al., 2012).

2) Our method is primarily based on theoretical modelling (Mie calculations for BC) and also includes empirical information on BC (e.g. size distributions, etc.). In contrast, the studies of Chung et al. (2012) and Bahadur et al. (2012) are purely empirical.

In Bahadur et al. (2012), the values of AAEBC and AAEOC in equation 1 above were determined from AERONET observations. They sorted the AERONET sites into regions by different sources. For example, they assumed the lower end of the AAE values reflects absorption due to BC. By averaging different fractions of the total frequency distribution at the different AERONET sites, they assume  $AAE_{440/675nm} \sim 0.55$  and  $AAE_{675/870nm} \sim 0.83$  for the pure BC condition. They then assumed  $AAE_{440/675nm} \sim 4.55$  for pure OC by using the AAE distributions. It is challenging to evaluate these thresholds numbers that were estimated from a few sites considering that the source of BrC is still under debate (Laskin et al., 2015). To summarize, Bahadur et al. (2012) (same for Chung et al. 2012) both used empirical information from select AERONET observations to derive the BrC absorption in a different set of AERONET observations.

In comparison, we do not apply such empirical information since conditions and properties can vary (e.g. the theoretical BC-AAE shown in our Fig.1). The BC-WDA-AAE relationship was calculated by Mie theory. Unlike Chung et al. (2012) and Bahadur et al. (2012), our method is based on the theoretical calculation using the information provided by the observation itself, not the empirical results from other absorption observations.

3) Because our method does not rely on any previous absorption observations, it is possible to apply it to observations other than AERONET. Our Figure 2 is only an example and our method is not restricted to the 440/675/870 wavelengths pairs. For example, we apply our method to surface measurements to derive BrC absorption at 370nm in Section 4. This is challenging for purely empirical method like Chung et al. (2012) and Bahadur et al. (2012) since there are limited previous observations of absorption at this wavelength.

In summary, our method is different from Bahadur et al. (2012) and Chung et al. (2012) from the initial idea to the applications. Given that 2 of 3 reviewers misunderstand our method, we have included a short discussion to describe the difference between our method and others (Page 6, line 22):

"A number of previous studies have used the AAE to separate the BC (or BrC) contribution from total absorption, however, these analyses typically rely on empirical information from previous observations. For example, Bahadur et al. (2012) and Chung et al. (2012) apply the same approach where they group AERONET sites by regions and possible source types and by analyzing these groups, they estimate the possible AAE (or SSA or EAE) and the corresponding range for pure BC or pure BrC. They then apply these empirical constraints to estimate the BC or BrC contributions at other sites. In contrast, our method uses the theoretical relationship between AAE and WDA for BC shown in Figure 2 in combination with the observed total AAE, and does not rely on any other data. This also makes our method "wavelength-flexible". Although we use the 440/675/870nm to describe our method, any three wavelengths with one in the near-UV and two at longer wavelengths in the visible spectrum can be used."

The authors go on to compute WDA for black carbon (BC) using a set of lognormal distributions (p4, lines 22-31), and provide a range of WDA values for their computations as the grey shaded area in their Figure 2. The median of this range of values is the black line, which represents the WDA\_0 of Eq 4. Thus, the authors are using theoretical BC AAE calculations as a baseline AAE; differences from this baseline are attributed to BrC. Particle size effects on AAE are not considered, although extreme dust cases are later omitted from the analysis (EAE < 1 and AAE > 1.5).

We must clarify that we are not using any BC-AAE as a baseline AAE. In our Equation 4 and Figure 2, BC-AAE is determined by not only theoretical WDA but also the measured AAE675/870nm. For different absorption measurements, different WDA and BC-AAE would be calculated. This method includes the effects of changing properties (e.g. size distribution) in the measurement itself.

In addition, the BrC absorption is not calculated using the BC-AAE from WDA0. It is the median value of the largest possible value (calculated from the lower end of BC-WDA range) and the lowest possible value (or zero, calculated from the higher end of BC-WDA range). We have clarified this point in the text (Page 5, line 30 – Page 6. line 2):

"We calculate the highest and lowest possible BrC absorption at 440nm based on the lowest and highest WDA (WDA1 and WDA2) as follows:

| $BC AAE_{440/870} = AAE_{675/870} + ln(WDA)$ | (3) |
|----------------------------------------------|-----|
|----------------------------------------------|-----|

BrC abs(440) = abs(440) - BC abs(440) (4)

The BrC absorption at 440nm is calculated as the median of these highest and lowest possible absorptions. For those points that fall within the shaded region, the BrC absorption is determined as the median of the highest possible absorption and 0."

There are many problems with this. First of all, the description of the parameters used for their Mie calculations does not include enough details for a reader to understand and duplicate the shaded region of figure 2. For instance, how thick are the coatings when the absorption enhancement factor is 2? What are the optical properties of the coatings? How does varying the optical properties of the coatings affect the size of the shaded area?

We have added a more detailed description of the coating calculation to the text (Page 4, line 26-30):

"The refractive index for coated material is assumed to be 1.55-0.001i, which is the typical value for non-absorbing organic and inorganic (Kopke et al., 1997). We first assume the coating thickness is 10% - 100% of the BC core radius and then only select the calculations with absorption enhancement smaller than a factor of 2."

**Which size distributions represent the baseline? Are these 20nm particles, or 300 nm particles? Do the baseline particles have coatings? How thick?**

As mentioned above, we do not have any baseline BC-AAE or BrC-AAE in our method. If reviewer means WDA0 here, it is the median BC under a range of size and coating conditions with a certain  $AAE_{675/870nm}$ . It does not represent a specific size distribution or coating condition.

For example, if  $AAE_{675/870nm} \sim 0.5$  is measured, we estimate WDA0 ~ 0.85 for BC from Figure 2. As shown in our Fig. 1, an  $AAE_{675/870nm}$  of 0.5 is consistent with BC larger than 100nm. Then the WDA0 reflects the mean of different size/coating conditions for these large particles. In summary, WDA0 depends on observation itself, and does not reflect a specific baseline condition.

**Which AERONET sites are falling inside of the shaded region of Fig 2?**

Figure 2 is intended as an illustration of our method and we previously showed one year of AERONET data. However, we have modified Figure 2 to correspond with data presented later in the manuscript. Now the points in Figure 2 are the seasonal mean values of 2004-2015 10 years averaged AERONET data in northern hemispheric winter (December, January and February). So the sites in Figure 2 are now the same as those in Figure 3a.

The sites falling inside of the shaded area are painted red in the figure below. This information is not particularly useful since these data points are calculated as 10 years seasonal means and our later analysis is based on the daily data. We have clarified this point at the beginning of section 3.

**Urban OC is non-absorbing, so the urban aerosols should fall inside of the shaded region, and the biomass burning aerosols should fall outside of the shaded regions. Looking at Figs 3&4, I don't believe that this is the case.**

The source of BrC is still under debate, it is still unclear that whether urban OC is absorbing or not (Laskin et al., 2015). BrC absorption has been observed in US background air (Liu et al., 2015). Furthermore, an urban site can also be impacted by BrC from other sources. Because the availability of AERONET AAOD is not good, many of the 10-years averaged values shown in Figures 3&4 reflect the mean of only several days. To ensure that our means are more representative, we now show only the sites with more than 6 years' measurements available in a single season. We also include a stricter threshold to exclude the data affected by dust. This is described in the paragraph at Page 5, line 15 (shown below). Figure 3 and 4, and the discussion in Section 3 are modified accordingly.

"While AERONET provides global observations of the column-integrated AOD, few of these sites actually have continuous measurements of AAOD throughout the year because the SSA is not always retrieved. For example, more than half of the AERONET sites measured AAOD for only 1 month in 2014. As a result, we use the data from the past decade (2005 – 2014) to enhance our sampling. To reduce the influence of sporadic events in the analysis, when showing the 10 years seasonal average value only sites with data for more than 6 years within a given season are selected. The AAOD from AERONET not only reflects the absorption from BC and BrC, but also that from dust. We use two thresholds to exclude the data possibly affected by dust. First, we use the coarse-mode AOD contribution (at 440nm) provided by AERONET. We assume that dust controls the total extinction of particles larger than 1 µm diameter (coarse-mode), and therefore remove data with a coarse-mode AOD contribution > 20% from our analysis. Second, the data exhibiting extinction Ångström exponent (EAE) < 1 are also considered to be influenced by dust and are removed prior to our analysis (Russel et al., 2010; Chuang et al., 2012)."

How come the shaded area of Figure 2 does not include the point (1,1), which would represent AAE = 1 for the entire wavelength range? Some of the particles sizes for their calculations are small enough (20 nm GMD) to reach the theoretical limit for small absorbing particles.

Thank you for pointing this out. Our Figure 2 should indeed include the point (1,1). Our original figure was smoothed across the intervals of AAE675/870nm. We have modified Figure 2 to eliminate this smoothing and more accurately present our results.

Why aren't any coarse mode aerosols included in the computations of the shaded region? There is always a coarse mode in the AERONET retrievals, and this will affect AAE. Sure, the authors eliminate aerosols dominated by dust in Section 3.1 by filtering for EAE < 1, but aerosol systems with 1 < EAE < 2 are mixtures of fine and coarse particles. Thus, there is dust in much (most?) of their analysis.

As described above and in the text, the shaded region in Fig. 2 is for BC only. Our goal here is to separate BC from other absorption. Therefore there is no issue associated with coarse mode aerosols in Fig.2.

We agree with the reviewer that, despite our filtering, the AERONET data may include some influence from dust. It is challenging to completely eliminate dust from these observations. We include a stricter threshold to exclude the data affected by dust. This is described in the paragraph at Page 5, line 15 as shown above.

However, we would like to point out that there is no such problem when applying our method to the fine particle absorption measurements made at the surface.

**How well is the AERONET AAOD known? How does the uncertainty in AAOD at each wavelength affect the uncertainty in the AAE in the two spectral regions that you are using? How do the AAE uncertainties affect the BrC AAOD retrieval?**

We have added a paragraph to discuss this point (Page 7, line 30):

"One challenge of this analysis is the well-known uncertainties associated with the AERONET observations. The measurement uncertainty is  $\pm 0.01$  for AOD,  $\pm 0.03$  for SSA when AOD > 0.2 and could be as large as  $\pm 0.07$  for SSA when AOD < 0.2. The uncertainty of AAOD depends on the corresponding AOD value, for example, this uncertainty is  $\pm 0.015$  with AOD = 0.4. Because our method is sensitive to the AAE not the AAOD, the uncertainty could be small for our BrC

contribution analysis if such uncertainties from AERONET are similar at all wavelengths. If the AERONET AAOD uncertainties vary substantially with wavelength, the influence on our analysis could be large and hard to quantify. In addition, the AERONET uncertainties suggest AAOD < 0.01 is certainly below the observed detection limit. In Figure 3 and in the above discussion most sites exhibit derived BrC-AAOD smaller than 0.01. However, all of these values are seasonal means over 10 years, and include both non-BrC detected (BrC-AAOD~0) data and BrC detected data. If instead we replace the BrC-AAOD < 0.01 data points by BrC-AAOD = 0.005 (the median of 0 and 0.01), the results are very similar."

Beyond the theoretical issues with this approach, though, the results do not seem to back up this approach. Figure 3 shows regional maps of seasonal mean BrC AAOD. The biomass burning sites in S. America never show elevated BrC AAOD, and the biomass burning seasons are not detectable over Africa. The Asian sites seem to have elevated BrC in all seasons except for Summer.

Figure 4 provides seasonal mean fractions of (BrC AAOD) / (total AAOD), and those results are even more ambiguous. The colors on these maps are all very random, with dark reds co-mingled with light blues; one would expect neighboring AERONET sites to exhibit similar colors if they were representing similar source regions. The randomness of the colors on these maps, though, seem to indicate noise.

As indicated above, the results of Figs 3&4 are impacted by the quality of AERONET data, which have been extensively discussed in literature. To address this issue, we now show only the sites with more than 6 years' measurements available in a single season. We also include a stricter threshold to exclude the data affected by dust. This is introduced in the paragraph at Page 5, line 15 as shown above. Figure 3, 4, 5 and the discussion in Section 3 are changed accordingly. We note, that this filtering does not significantly impact the distributions shown in Figure 6. We believe that Figures 3 and 4 now better represents salient geographical features, however we note that the biomass burning seasons still exhibits low BrC-AAOD over Africa. This is because such sites/data are removed from the analysis when we filter for dust.

**OTHER IMPORTANT POINTS:**

**Page 3, line 6: AERONET provides \*retrievals\*, not measurements.**

Changed.

**Page 3, line 23-25: Chung (2012) and Bahadur (2012) do not assume AAE = 1 for BC, as stated here. The authors need to be more careful when attributing concepts to others.**

Here we meant that Chung et al. (2012) and Bahadur et al. (2012) also pointed out the problems associated with assuming AAE = 1 for BC. We did not mean to imply that they assumed BC-AAE = 1 in their studies. We have deleted them from the references here to clarify this point.

**Page 5, line 8: Bahadur (2012) has nothing to do with the SSA screening conditions of AERONET. Cite Dubovik (JGR, 2000), instead.**

Changed.

**Page 5, line 16: The authors state: "... and cannot be estimated using our method without additional information about the size and coating state of the particles." However, you have information about size in the AERONET retrievals. Why not use them?**

The sentence has changed to "... and cannot be estimated using our method without additional information about the size and coating state of BC particles."

The size distribution from AERONET retrievals reflects the entire aerosol in the column which always contains a combination of constituents. However, it is challenging to estimate the BC size distribution from this total size distribution.

**Page 5, equations 3&4: These equations should be reversed. WDA\_0 is used to compute BC AAE\_440/870, which is used to compute BC abs(440), right? So presently, Eq 4 gets used before Eq3.**

The reviewer is right, we have reversed the sequence between equation 3 and 4.

Page 6, line 18: The authors state: "In general, BrC-AAOD is smaller than 0.005 at most sites but larger in Asia and occasionally in northern Africa and South America." BrC-AAOD < 0.005 in general? That sounds like a small number. Does the AERONET retrieval really have the sensitivity to quantify this parameter, then? AERONET only claims an accuracy of \_0.01 for AOD; the accuracy of AAOD is certainly no better. So all of the BrC-AAODs that you are retrieving seem to be well below AERONET's accuracy threshold.

Note that all the values shown in Section 3 are 10-year seasonal averages, which include days of both with and without BrC. The data with ~0 BrC-AAOD bring down the average. As shown

above, we have added new text to address the uncertainties of AERONET data in paragraph of page 7, starting at line 30.

Page 6, lines 24-26: The authors are quoting numbers for seasonal variability, but they do not specify seasons or sites. The colors on the maps do not obviously correspond to the numbers quoted here. The authors should at least give these "special" sites a different shape, so that the reader can understand which sites are the "African biomass burning sites." Also, the biomass burning season on the Sahel is not July-October. It starts in November and ends in the Spring.

As indicated above, we now use new thresholds to select AERONET data. This part of discussion has been changed as well as Figure 3.

Page 6, line 31: The authors state: "In winter the BrC AAOD contributions in North America (15%), Europe (17%) and East Asia (23%) are larger than in summer (10%, 13% and 14%)." Why should BrC AAOD fractions be higher in the winter in North America? Few Americans use biofuel, and the biomass burning season is in the summer. Also, higher winter BrC AAOD fractions are not obvious from Fig 4.

As indicated above, we now use new thresholds to select AERONET data. This part of discussion has been changed as well as Figure 4.

**Equation 6: The authors assume that all OC are BrC for Equation 6. This is not a valid assumption.**

Yes, in principle we agree. However, as we do not know what fraction of OC is BrC, nor do we have measurements of BrC mass, it is easiest to estimate the MAC for all OC. As a result, we analyze the optical properties of total OC as a combination of both BrC fraction and BrC optical properties. This is discussed at the end of section 3.2.

Page 12, line 10: The authors state "the correlations between BrC-AAE and BrC absorption contributions are only slight to moderate at these sites." They should still provide the correlations to the readers, though.

Added.

Page 12, lines 22-23. The authors state that OA-MAC is correlated with BC/OC mass ratio, but they do not help the reader to understand why this should be the case (anywhere in the article). Thus, for a given OC mass, the OA-MAC is linearly related to BC mass? This sounds like BC is contaminating their OC-MAC retrieval (rather than proof that their retrieval is working).

This relationship has been shown in the literature (e.g. Saleh et al., 2014) and is likely to relate to the combustion efficiency. We discuss this in section 3.2 and section 5.

**Page 13, lines 26-28 and Figs 10b, 10c: Authors find that the OA-MAC for the nonbiomass plumes (Fig 10b) are much much higher than for the BB plumes (Fig 10c). WHY? OA-MAC for urban aerosols should be negligible.**

Given the present poor understanding of BrC, it is not clear that OA-MAC is negligible under all urban conditions. OA-MAC is determined not only by OA absorption but also the OA mass. A small OA absorption with high OA-MAC is possible.

A possible explanation for this relates to the BC/OA ratio. As discussed in the text, OA-MAC is found to be positively correlated with BC/OA emission ratio (Saleh et al., 2014). The combustion sources in urban regions typically have much larger BC/OA emission ratios than biomass burning, which could result in higher OA-MAC.

**Reviewer #2**

This paper attempts to quantify brown carbon absorption using the AERONET data base of aerosol extinction optical depth and single scattering albedo as well as aethalometer measurements at a number of stations. The approach used in this analysis has been used in previous studies [Schuster et al., 2005; Russell et al., 2010, to mention a few], strangely ignored in the literature review. An innovative component of this work is the combined use of surface and space based AAOD retrievals for the estimation of aerosol absorption exponent. There are several issues in the paper that need to be addressed.

We respectfully disagree with the reviewer that our approach has been used in previous studies. The methods in Schuster et al., 2005 and Russell et al., 2010 are very different from ours.

The core idea of our method is to find the range of BC absorption in any multiple-wavelength absorption measurements. This is estimated based on the difference of AAE at different wavelengths pairs (WDA). Our method is primarily based on theoretical modelling (Mie calculations for BC) and does not rely on any information from previous observations. The most important result of our method is the theoretical relationship of BC-WDA and BC-AAE shown

as the shaded region in Figure 2, which, to our knowledge, has not been discussed in any previous studies. Furthermore, our method is "wavelengths-flexible". It is possible to apply it to the observation from not only AERONET but also absorption measurements at other wavelengths.

The work of Schuster et al., 2005 focuses on a completely different topic from this work. They use the Maxwell Garnett effective medium approximation to infer the black carbon concentration and specific absorption of AERONET retrievals. They focus on the AERONET data at 550nm and do not use any information from either the AAE or the WDA. In their analysis, they use a fixed black carbon refractive index of 2-1i and an ammonium sulfate refractive index of 1.53-10-7 i at 550nm as well as the size distribution of total aerosols provided by the AERONET retrieval. They do not consider BrC as a contributor to absorption, though they estimate the uncertainty from organics by referencing other measurements.

Russell et al., 2010 try to use AAE and EAE to identify the aerosol composition measurements and discuss its future application. By analyzing 11 non-oceanic AERONET sites and other published observations, they found that the AAE values are strongly correlated with aerosol composition. For example, AAE near 1 is dominated by urban-industrial aerosol, larger AAE values are dominated by biomass burning aerosols, and the largest AAE values are indicative of dust aerosols. Their purpose and methods are very different from ours. First, they want to convey the idea of connecting AAE with aerosol composition, but do not attempt to separate BC from BrC as we do. Second, their analysis is based on previous observations. Third, although they use the information from the AAE, they do not address the wavelength dependence of AAE (WDA), which is the core factor in our method.

Since 2 of the 3 reviewers misunderstand our method, we have added a short discussion to the text to describe the difference between our method and others (Page 6, line 22):

"A number of previous studies have used the AAE to separate the BC (or BrC) contribution from total absorption, however, these analyses typically rely on empirical information from previous observations. For example, Bahadur et al. (2012) and Chung et al. (2012) apply the same approach where they group AERONET sites by regions and possible source types and by analyzing these groups, they estimate the possible AAE (or SSA or EAE) and the corresponding range for pure BC or pure BrC. They then apply these empirical constraints to estimate the BC or BrC contributions at other sites. In contrast, our method uses the theoretical relationship between AAE and WDA for BC shown in Figure 2 in combination with the observed total AAE, and does not rely on any other data. This also makes our method "wavelength-flexible". Although we use the 440/675/870nm to describe our method, any three wavelengths with one in the near-UV and two at longer wavelengths in the visible spectrum can be used." The use of annual average AAOD values in Figure 2, ignores the seasonal variability of aerosol type. It would be more illustrative to plot instead seasonally averaged data and identify in a table the sites used in the analysis. I suspect the analysis presented in section 2 will change substantially if seasonal AAOD data is used.

We have changed Figure 2 to show the seasonal averaged data. These points now also correspond to those shown in Figure 3a. However, we would like to mention that the purpose of Figure 2 is primarily to illustrate our method.

The seasonal maps showing the 10-year average of seasonal mean BrC-AAOD (as indicated in the caption of Figure 3) look very strange to me. I wonder if that is really what is being plotted. I am very surprised by the number of AERONET sites having 10 years of observations over the 2005-2014 period. Most sites shown on those maps do not have a continuous ten year record. Surprisingly the seasonal maps of derived BrC-AAOD based on AERONET observations do not show the expected AAOD hot spots in the Sahel (DJF), Southern Africa (JJA, SON), and Amazon Basin (SON). The only known hot-spot correctly showing is the Southeast Asia (MAM). I suspect these maps are more a representation of the geographical and temporal bias of AERONET observations over the 2005-2014 period than an actual representation of the AAOD load as intended by the authors. The number of sites in SE Asia and China has increased significantly over the last ten years. On the other hand, the coverage over Southern Africa has gone down significantly. A better description and explanation of this map is needed. I suggest to re-do these maps using a more careful selection of sites making sure that the temporal coverage is similar for all used stations.

Thank you for raising this issue. We have modified our thresholds to be more selective with the AERONET data. Only sites with > 6 years measurements available are used in the analysis. This is described now in the paragraph at page 5, starting at line 15. The poor coverage over Africa is because data that is influenced by dust is removed by the analysis; we comment on this on page 7, line 12:

"In contrast, no significant seasonal variations are found in other regions. The sites in Africa exhibit low BrC-AAOD even during biomass burning seasons. This is because nearly all the data with high AAOD in Africa are excluded from the analysis due to the influence of dust."

**Other comments:**

Page 6, line 24. Abundant biomass burning also takes place in Northern South-America and the Sahel during NH winter. Also in the NH Spring there is intense biomass burning in Southeast Asia and Central America.

We have changed this part of discussion based on our new AERONET data selection. In our analysis, the influence of biomass burning in South America and Southeast Asia are significant. We agree with the reviewer that the Sahel during NH winter and Central America spring are also characterized by abundant biomass burning, however we do not have enough data points in these regions to characterize this. We comment on this on page 7, line 12 as shown above.

**Page 9, line 1. Add the Torres et al [2014] reference that describes the latest algorithm upgrades.**

Added.

Page 9, line 8. Nowhere in the quoted references, is it said that OMI and AERONET AAOD have a correlation > 0.8 as stated by the authors. They incorrectly imply that OMI and AERONET AAOD have been directly compared citing papers that refer to AOD validation. Both OMI AOD [Ahn et al. 2014] and SSA [Jethva et al, 2014] have been compared to AERONET produced values. The authors need to read the papers and correctly quote them.

The reviewer is correct, it should be "AOD". This has been clarified in the text now.

The title of the paper is confusing because both AERONET and aethalometer data are obtained by means of surface observations. The title should also reflect the fact that OMI observations are used in the estimation of the reported results. Suggest changing title to 'Deriving brown carbon absorption from multi-wavelength absorption measurements: Method and applications to AERONET, OMI and Aethalometer observations'

Thank you for raising this point. We agree that the "aethalometer" should be used instead of "surface". However, we do not think it is appropriate to include OMI in the title. We use OMI to complement the analysis of the AERONET observations, but do not derive brown carbon from OMI. The title has changed to "Deriving brown carbon absorption from multi-wavelength absorption measurements: Method and applications to AERONET and Aethalometer observations"

**Reviewer #3**

This work provides an observational constraint for brown carbon aerosols (BrC), which absorb solar radiation, and thus have an important implication for climate. Based on the previous study for quantifying the brown carbon aerosol absorption, authors used observed absorption angstrom exponents at a pair of wavelengths at AERONET sites to reduce associated uncertainties. I believe that this dataset adds up to the observations of BrC absorptions, which are very sparse globally and thus it will be valuable to evaluate the estimated contributions of BrC to aerosol absorption and radiative forcing. However, the observed quantity derived from the combination of various observations needs additional clarification and the details are listed as follows.

P4, L23 - Authors need to estimate the associated uncertainty with the assumption of spherical BC in their method. For example, Kahnert and Devasthale et al. (2011) estimated the difference of SSA up to 0.05 between spherical vs. aggregate shapes of BC (Figure 5 in their paper).

This is a good point. We have added a short discussion to mention this issue (Page 6, line 18-21) as shown below. However, we do are unable to estimate the uncertainty since there is little information on how aerosol shape might affect AAE and WDA.

"The spherical assumption in the Mie calculations could lead to additional uncertainties, as previous work suggests that the shape of BC can affect both the SSA and the absorption enhancement from coating (Adachi et al., 2010; Kahnert and Devasthale, 2011). However this uncertainty is hard to estimate since it is difficult to quantify how particle shape influences AAE and WDA."

**P5, L21-24 - Please elaborate how you obtain 4% uncertainty.**

In the text, this 4% is the detection limit, not the uncertainty. We assume the reviewer means the detection limit here.

As described in that paragraph, we estimate the uncertainty of BC absorption by repeating the calculation using the lowest and highest WDA value of the shaded region in Figure 2. The uncertainty of BrC absorption depends on both BC absorption uncertainty and the contribution of BrC to total absorption. If the uncertainty is equal or larger than the BrC absorption, the BrC absorption is below the detection limit. The minimum uncertainty for BC absorption is estimated to be ~4% (based on the smallest variability of WDA). Then we could assume a series of BrC absorption contributions. For example, when BrC contribute 50% of the total absorption, the uncertainty is estimated to be 4.2% based on the minimum ~4% uncertainty for BC. Assuming the total absorption is  $\tau$ , then the real BrC absorption is 0.5 $\tau$  and the uncertainty is 0.5x4.2% = 0.021 $\tau$ . As the BrC contribution decreases, the real BrC absorption decreases. When BrC absorption contribution > 4%, the real BrC absorption is larger than the uncertainty; when BrC absorption contribution < 4%, the real BrC absorption is lower than the uncertainty, thus below the detection limit.

**Figure 2 - It would be recommended to remove the dust contribution as shown in Section3.**

We have changed Figure 2 to show the 10 years seasonal averaged data for winter instead of the annual averaged data for 2014. The data points are filtered for dust as described in the paragraph on page 5, line 15:

"While AERONET provides global observations of the column-integrated AOD, few of these sites actually have continuous measurements of AAOD throughout the year because the SSA is not always retrieved. For example, more than half of the AERONET sites measured AAOD for only 1 month in 2014. As a result, we use the data from the past decade (2005 – 2014) to enhance our sampling. To reduce the influence of sporadic events in the analysis, when showing the 10 years seasonal average value only sites with data for more than 6 years within a given season are selected. The AAOD from AERONET not only reflects the absorption from BC and BrC, but also that from dust. We use two thresholds to exclude the data possibly affected by dust. First, we use the coarse-mode AOD contribution (at 440nm) provided by AERONET. We assume that dust controls the total extinction of particles larger than 1 µm diameter (coarse-mode), and therefore remove data with a coarse-mode AOD contribution > 20% from our analysis. Second, the data exhibiting extinction Ångström exponent (EAE) < 1 are also considered to be influenced by dust and are removed prior to our analysis (Russel et al., 2010; Chuang et al., 2012). "

**P7, L14-15 - Several papers showed that BrC absorption at 675 nm is significant (Alexander et al., 2008; Chung et al., 2012). So I am wondering if you assume an absorption at 879 nm as BC absorption alone, then how would your results differ. Or at least, you many need to discuss those previous papers and the possible effect on your estimates.**

Thank you for raising this point. The assumption in our method is that BC dominates the absorption at 675nm regardless of the BrC absorption value. Neither Alexander et al., 2008 nor Chung et al., 2012 show the BC absorption contribution at 675nm. As mentioned in the text, we need at least two wavelengths measurements with negligible BrC contributions. Therefore our method would not work if we only assumed that BrC was negligible at the one wavelength (879 nm). We have added such discussion in the paragraph at page 8, line 7:

"In our method, we assume that the BrC absorption contribution is negligible at 675nm and 870nm. This is supported by the laboratory measurements (Chen and Bond, 2010; Zhang et al., 2013; Yang et al., 2009; Kirchstetter et al., 2004). However, Alexander et al. (2008) find the BrC absorption may be significant at 675nm by examining an electron energy-loss spectrum from a transmission electron microscope. If BrC absorbs significantly at 675nm, our estimate of BrC absorption at 440nm would be underestimated."

**P7, L19 - Why did you use GFED3 for 2011 and earlier? If there is no reason for this, you better use GFED4 for the entire period consistently.**

We now use GFED4 for the entire period.

**P9, L8-9 - Jethva and Torres (2011) and Ahn et al. (2014) conducted an evaluation of AOD alone, not AAOD.**

This should be AOD, we have clarified this in the text.

Jethva et al. (2014) did an SSA evaluation and showed that OMI SSA are higher than AERONET SSA. For example, about 50% of total samples showed the difference of 0.03 or higher and 25% showed 0.05 or higher differences. This is a considerable discrepancy between two datasets and may have a huge impact on your estimates. For example, if AERONET SSA is 0.94 and OMI is 0.97, then the estimated AAODs using AERONET versus OMI data differ by 100%. So BrC AAE using OMI and AERONET AAOD together may cause too high uncertainty. Please consider a bias correction for OMI or simply drop out OMI data in your calculation. minor corrections,

Jethva et al. (2014) shows that reasonable agreement is found between the two techniques globally, especially for the "urban/industry" type retrievals. Globally, 49% data at low AOD condition (AOD < 0.7) and 53% data at high AOD condition (AOD > 0.07) fall within the 0.03 uncertainty limit. OMAERUV SSA is either higher or lower than AERONET SSA at different site, not always higher as the reviewer stated. Significant differences between the two datasets are mostly shown at dust-dominated sites, which are excluded in our analysis. In addition, this comparison was made at 440nm. The OMI retrieval at this wavelength is more uncertain than the 388nm data we use in the analysis. It is not possible to directly compare the SSA/AAOD at 388nm since AERONET does not make measurements at this wavelength. Therefore we believe that the comparison between AERONET and OMI is still valid. We have added a discussion on this to the paragraph at page 10, line 31:

"Many studies have evaluated OMAERUV AOD by comparing them with ground-based measurements. The correlation between OMAERUV and AERONET AOD is usually found to be high (R > 0.8) (Jethva and Torres, 2011; Ahn et al., 2014). Jethva et al. (2014) also compare the SSA between OMAERUV and AERONET and find 69% of the data agree within the absolute difference of ±0.05 for all aerosol types. Significant differences between the two datasets are most shown at dust dominated sites. These dust-influenced sites are not included in our analysis. Furthermore, Jethva et al. (2014) compare these products at 440nm. The OMAERUV SSA estimated at this wavelength relies on a number of assumptions and is more uncertain than that reported at 388nm that we use in our analysis. It is not possible to directly compare the SSA/AAOD at 388nm since AERONET does not make measurements at this wavelength. Therefore we believe that the comparison between AERONET and OMAERUV is still valid. . However, if the OMAERUV SSA is higher or lower than AERONET at 388nm, our estimate of the BrC-AAE388/440nm would be biased low or high."

**P2, L28: Forrister et al., (2015) -> Forrister et al. (2015) P7, L15: 675 m -> 675 nm**

Changed.

---

## Author Comment (AC3) · 6 Jun 2016

The comment was uploaded in the form of a supplement:
http://www.atmos-chem-phys-discuss.net/acp-2016-237/acp-2016-237-AC3-supplement.pdf
* * *

---

## Author Response (AR2)

Dear Editor,

Thank you for the opportunity to address the concerns raised in review and in the Editor's report.

There are 3 main components to our manuscript: (1) the development of a new AAE methodology for estimating BrC absorption from multiple-wavelength absorption measurements (2) application of this method to AERONET observations (3) application of this method to 10 aethalometer datasets and an exploration of BrC aging. This paper presents new constraints on the fraction of BrC absorption world-wide (while observationally bracketing the BrC AAE), as well as novel insight into the aging of BrC from urban and biomass burning sources. We believe that the bulk of the concerns raised in review are related to confusion regarding the methodology and the application of our methodology to the AERONET data, and have addressed these in detail in the revised manuscript.

Please find below responses to the re-review of Reviewer #1, responses to the Editor's comments, and a general commentary on previous misinterpretation of our new methodology, as you have requested. Please let us know if we can provide any further clarification.

Sincerely,

Xuan Wang, Colette Heald, Arthur Sedlacek, Suzane de Sa, Scot Martin, Lizabeth Alexander, Thomas Watson, Allison Aiken, Stephen Springston, and Paulo Artaxo

**1. COMMENTS FROM REVIEWER #1**

*Note: page numbers are for initial Response to Reviews from authors*

Page 2:

I still don't understand why the authors define WDA = exp(AAE_440/870 - AAE_675-870) and then always use the natural log in their equations. That way, Eq 3 becomes $BC_{?}AAE440/870 = AAE675/870$ - (AAE_440/870 - AAE_675-870). Thus just makes things more complicated than needed.

We now define WDA = $AAE_{440/870}$ - $AAE_{675-870}$.

Page 4 (bottom):

Well, this information is useful because it tells the reader which sites have BC but very little BrC, right? Thus, one would expect urban areas to dominate the grey areas. This does not seem to be the case in the figure below.

The red sites simply indicate where BrC cannot be detected by our method, this may result from either negligible contributions of BrC, or overall low absorption. While the former case may be consistent with urban regions (e.g. sites in the East Coast of the US), this may not be the case for the later. We also note that urban sites may also be characterized by residential burning sources of BrC. Recent field measurements also suggest that BrC absorption does not contribute absorption in urban area (Liu et al., 2015).

Furthermore, as we already mentioned, this information is not particularly useful since the data points are calculated as 10 years seasonal means and our later analysis is based on the daily data. The red points do not mean BrC cannot be detected every day during 10 years at the site. We use this seasonal figure to answer the reviewer's question because it is not reasonable to show all the daily results during 10 years. As we clarified at the beginning of section 3, Figure 2 is only for illustration and is not directly related to the later analysis.

Page 5:

So the authors contend that the urban aerosol BrC signal is as strong as the BB BrC signal? Then this should be part of the discussion, because many readers believe BrC to be dominated by bioburning or hulis sources. The point is that seasonal and regional trends that you deduce should be consistent with the literature.

Given the sparse AERONET data record, we do not believe that the observational constraints are sufficient to make such broad conclusions. We agree with the reviewer that biomass burning is thought to be a major source of BrC, and as discussed on page 7 of the manuscript, we do estimate higher BrC-AAOD in South America and Southeast Asia during the biomass burning season, in line with this assumption. For Africa, nearly all the data with high AAOD are excluded from our strict exclusion of

dust as discussed in section 3.1. We also see from aethalometer data in Figure 7+8 that BrC absorption is high in Europe, a biofuel-dominated region.

In addition, as we show in Figure 10c, that it appears that BrC from biomass burning may photochemically whiten in the atmosphere (consistent with Forrister et al., 2015). We see no evidence for such whitening from urban aerosol (Figure 10b). This distinct difference in aging can only be assessed for the one site where we have sufficient observational constraints to support this conclusions; however, it may suggest one possible explanation for why biomass burning regions are not more prominent sources of BrC in Figure 4.

We have added some discussion of this to Section 3.1 and the Discussion.

PAge 6 (middle):

This is irrelevant for AERONET analyses. The imaginary refractive index of the coarse mode is the same as the imaginary index of the fine mode in the AERONET database. Always! Thus, there is always some coarse mode absorption; oftentimes, there is more absorption than can be rationalized by assuming 100% dust in the coarse mode.

We agree with the reviewer regarding the refractive index applied to both the fine and coarse model of AERONET observations. We also agree that signatures of dust are challenging to remove from the AERONET observations. To address this we previously implemented both a coarse mode filtering, to remove coarse mode dust, as well as an EAE > 1 criteria for dust filtering as suggested by previous studies (Russell et al., 2010; Chuang et al., 2012). We now further strengthen this by increasing our coarse mode filtering, and applying the filtering described by Bahadur et al. (2012) to select only SAE>1.2 or use AAE ratios to obtain "dust-free" AERONET measurements. Bahadur et al. (2012) also indicate that fine dust absorption is negligible in the AERONET observations (top of page 17369 of their paper). While it is impossible to perfectly remove all signatures of dust from the AERONET observations, we believe that our strict filtering dramatically reduces the locations and times where AERONET data is contaminated by dust, and that our long-term averages are not significantly affected by this. The consistency between the BrC assessment from AERONET and aethalometer data  also suggests that dust contamination is not significant in our AERONET analysis. However, we

acknowledge that the AERONET observations are not a direct measurement of absorption, but a retrieval, and thus retrieval assumptions inherent to these measurements may impact the analysis.

To address the reviewer (and Editor)'s concerns we have implemented and described the Bahadur et al. (2012) dust-free filtering criteria to the text, we have acknowledged the concerns regarding AERONET retrievals, and we have added extensive caveats to the introduction and conclusions of our AERONET analysis.

Page 6, bottom:

You need to quote the source for these numbers so that others can follow up.

Done.

**2. COMMENTS FROM EDITOR**

AERONET assumes internally mixed aerosols, with the exact same composition for both the fine and the coarse mode. "Always!", as the reviewer states, regardless of what happens in the real atmosphere. This means that there is always some coarse mode absorption, which might not be negligible, and the absorption from any constituent (BC and dust) are always in volume mixing with the scattering aerosols. By assuming different components, the manuscript assumes that when the "better" parametrization with externally mixed aerosols deviates from the "not so good" AERONET retrieval with internally mixed aerosols, indicates the presence of BrC. The big problem here is that the internally mixed aerosols do not behave the same way with the assumed externally mixed aerosols when BC/BrC explicit optical properties exist, neither when a core-shell assumption is made, when it comes to optics calculations. For this, the explicit modeling of absorption presented in the manuscript is inconsistent with the AERONET assumptions, so one can't directly compare the two, not even qualitatively. This is especially the case for AAE, which is very sensitive to the coarse mode volume fraction, even when fine mode dominates, which is not considered in the manuscript.

First, we would point out that our dust filtering criteria is now quite strict and is in line with the "dust-free" data category described by Bahadur et al.(2012) and therefore we feel that dust contamination is virtually eliminated in our analysis (see above). More generally we have strengthened our filtering of

coarse mode aerosol, such that the concerns about mixing of fine and coarse mode and AAE sensitivity to the coarse mode are alleviated. We also note that our analysis is consistent with previous analyses of AERONET observations which view the AAOD of different species to be an extensive and additive property (e.g. equation 3 of Bahadur et al., 2012). However, our estimated range of BC absorption and wavelength dependence includes the possibility of both pure and coated BC, thus allowing for internal mixing of BC and scattering aerosol.

We also would like to point out that though our filtering of coarse mode aerosols includes the mixing assumption from AERONET, our BrC analysis does not. After the data filtering, we only use AOD and SSA from AERONET retrievals in our analysis. These values are derived by solving the radiative transfer equation for a plane-parallel atmosphere, which do not include any assumptions related to the mixing state of aerosols. Although AERONET assume internal mixing for other retrievals (refractive index, size distribution, etc.), these retrievals are not included in our comparisons.

Finally, we do acknowledge that the AERONET retrieval is an indirect measurement, and retrieval assumptions may impact the reported absorption values. We have therefore added extensive caveats to the text to reflect concerns raised about this issue. Given the limited number of direct measurements of multiple-wavelength (which we note are not affected by these issues and are analyzed in Section 4), we include AERONET observations in our analysis despite data limitations, in order to provide at least a qualitative global constraint on BrC.

On a positive side, I believe the work presented is really worth pursuing a bit further. Brown carbon emerged to be an important aerosol component, and our understanding of it is really poor, if not worse. I would very much like to see a paper that you work with someone from AERONET, use your assumptions and repeat the AERONET retrievals, so create your own product, and then be able to say quantitatively something about BrC. Repeating the AERONET retrievals by paying attention to internal/external mixing and fine/coarse mode absorption (by using different refractive indices or different size assumptions) will allow you to compare apples with apples and make a study that I

believe will be very highly cited in the future. I would strongly recommend you to try to address this weakness and resubmit the manuscript for publication.

Thank you for your encouragement. While we believe that generating a new AERONET retrieval product is outside the scope of this work, we hope that you will agree that our extended discussion of the limitations of how our method can be applied to AERONET observations addresses the key concern raised by yourself and the Reviewer.

**3. COMMENTS ON METHODOLOGY**

In the initial review process, Reviewers #1 and #2 confused our approach with pre-existing methodology from the literature. Reviewer #2 suggests that our work is similar to Schuster et al. (2005) and Russell et al. (2010), however neither of these studies attempts to estimate BrC absorption or properties. Reviewer #1 suggests that our work is similar to Chung et al. (2012) and Bahadur et al. (2012), and while our method for estimating BrC relies on the wavelength dependence of absorption, similar to these two studies, it also differs in a number of ways; these are described extensively in our initial response to reviews and are summarized here. These two studies use assumptions about BrC and BC properties or estimate these from one set of observations and then apply them to another set of observations. In contrast, we use theoretical modeling (Mie calculations) to estimate the possible range of BC wavelength-dependent absorption (considering a range of coating states and sizes). From a set of multi-wavelength absorption measurements we can then define BrC as the near-UV absorption residual above the BC range. This method is illustrated by Figure 2. Our method provides several key advantages over previous work: (1) it does not require any a priori assumptions about the properties of BrC (2) it does not require a "training set" of observations over a given region (with assumed source influences) (3) it can be applied to any set of wavelength-dependent absorption measurements (not limited to AERONET as in previous studies due to the use of a "training set"), including the aethalometer data presented in Section 4 of our manuscript.

In responding to reviews, we acknowledged that this confusion may have arisen, in part, from an insufficient description and comparison of our approach with previous approaches in the literature. To address this we have added the following to the manuscript:

1. In Section #1, after describing previous studies which have assumed BC-AAE=1 to extract BrC, we introduce the idea of empirically assuming BrC-AAE and BC-AAE as done in previous studies (Chung et al., 2012; Bahadur et al, 2012).

2. At the end of Section #2 a paragraph describing previous approaches and comparing these to our method has been added.

[revised manuscript text omitted]

---

## Author Response (AR3)

**Response to Anonymous Reviewer #4**

Following is my general assessment of the manuscript and the anonymous reviews. There are two main issues with this paper that every reviewer touched on in some way:

1) The approach is essentially based on (or a form of) a well known method for predicting BrC that involves comparing observed absorption Angstrom exponents (AAE) to what is predicted for black carbon. This paper may add an improvement to this method, but basically all the well documented fundamental issues with the method as a way to calculate BrC remain and are a major source of uncertainty.

We agree with the reviewer that calculating BrC by comparing observed AAE to what is predicted for BC is a previously established approach (the so-called "AAE-method"). We also agree that such approaches can be quite uncertain, however in an era with extremely limited direct constraints on brown carbon, such indirect approaches, though uncertain, are tremendously valuable. As suggested by the referee, in our study we have improved the AAE-method and these improvements do decrease the uncertainties in many aspects:

The key question for this method is whether the assumed properties of BC are correct in the analysis. In previous studies, the AAE and absorption of BC have been predicted from either (1) empirical summary of other observations (e.g. Chung et al., 2012) or (2) using a purely theoretical fixed number (e.g. AAE=1 for BC) (e.g. Clarke et al., 2007). We discuss in the paper how the latter assumption (fixed BC AAE) has been shown in the literature not to represent true conditions of the atmosphere. The former approach is also limiting in that it assumes that BC can be separated cleanly in particular locations (questionable) and that the estimated BC properties for that location are then generally applicable to all other locations. In our method, we identify a range of assumptions for BC based on Mie calculations given the uncertainty on BC properties (size distribution and coating thickness). Using the information on the wavelength dependence of absorption provided by the measurements themselves, we are then able to estimate the BrC absorption, and thus decrease the uncertainty on this estimate.

The reviewer also refers here to "well-documented fundamental issues" regarding the AAE-method. We would like to point out that the fundamental AAE approach is sound and we are not aware of any studies which say otherwise (i.e. if one knows the wavelength-dependent absorption of BC, from a set of measurements of absorption at several wavelengths, one can estimate the BrC absorption. This is simple algebra.). The challenge is knowing what the BC contribution is, and as stated above, we feel that our approach offers an improvement over previous attempts.

In addition to the methodological uncertainty that was discussed in the 6$^{th}$ paragraph in Section 2, we have clarified the need for indirect approaches to estimate BrC (page 3, lines 3-7) and how we build on

previous approaches (page 3, lines 28-30).We have also re-organized the discussion of previous AAE-based methods in Section 2 to clarify how our approach differs from and builds upon previous work.

2) Use of AERONET data to infer BrC. Again, there are a number of publications detailing the major limitations with this approach.

We agree that the accuracy of our estimated BrC absorption from AERONET are challenging to estimate and may be large (we acknowledged this in Section 3.1 paragraph 1), given the uncertainty on the AERONET AAOD measurements. However, AERONET is the only global data set that can be used for such analysis and thus, at least qualitatively we believe that this analysis provides much-needed insight into global distribution of BrC. We also highlight that AAE-methods have previously been applied to AERONET data (e.g. Bahadur et al., 2012), and thus our approach is consistent with the peer-reviewed literature on this topic. While ideally our method would be applied to a more certain set of absorption measurements (as is done with the aethalometer data in Section 4!), there is value in the application to AERONET both in terms of the global distribution, and comparison with previous AERONET-based estimates. Finally, we note that from the AERONET analysis, in the abstract and conclusions we emphasize the overall relative contributions of BrC to absorption, not the specific absolute values of AAOD that we estimate.

It can be argued that the authors of this manuscript did not do a sufficient job of addressing both these well documented issues up front in a clear and concise manner, hence the consistent responses from the reviewer's. Furthermore, from a larger perspective, I think one could also argue that even though there is an understandable motivation to predict BrC global radiative impacts, the data to do this accurately doesn't exist at this point, making the efficacy of the increasing number of papers utilizing AAE approaches, like this one, questionable.

It appears here that the reviewer is referring to our original manuscript, not the edited version provided in response to review (there was only one re-review, and the comment above refers to multiple reviewers). We agree that our original manuscript did not sufficiently address these issues. Based on the concerns raised in the first round of review, we clarified these issues as well as how our work connected with previous studies. In this response, we have further edited the text (as described above) to address these concerns. We agree that better observational constraints on BrC are desperately needed. However, given the importance of BrC to global radiative forcing, the motivation to extract information from existing measurements is high. The reviewer's comments focus on our application of the AAE-method to the AERONET data, and indeed this application is uncertain (as we acknowledge) as it combines the uncertainty of the method with the high uncertainty from the dataset. However, we also applied our method to 8 aethalometer datasets, which do not suffer from the same issues as the AERONET data. The reviewer did not comment on this analysis (nor was it discussed in the previous reviews). The

analysis of the aethalometer data, particularly the application to the GoAmazon observations provide some of the key results in the manuscript. Finally, the limitations of presently available data do not diminish the scientific value of our method. Any future absorption measurements at at least three wavelengths with one in the near-UV and two at longer wavelengths in the visible spectrum can be immediately used to strengthen the analysis. We hope that this manuscript will motivate development of multi-wavelength measurement approaches that can be used in such an analysis. We have added a few sentences to emphasize these points in the last paragraph of Section 5.

**Comments from co-editor decision:**

I did request one additional review, from a new reviewer who is a well known expert on brown carbon, asking to explicitly comment on both the paper and the past reviews. As you can see from their report, the paper has strong weaknesses, and the new reviewer agrees with the past critical reviews. The interesting point the reviewer makes though is that "the data to do this accurately doesn't exist at this point, making the efficacy of the increasing number of papers utilizing AAE approaches, like this one, questionable". This means that based on their opinion there is no way to improve the results of the manuscript, due to limitations of the currently available data used as input.

As stated above in our response to reviewer#4, we do not agree that the "limitations of the currently available data used as input" diminishes the scientific value of our manuscript:

1. Although the uncertainty of AERONET absorption measurements make it challenging to estimate the accuracy of absorption numbers in Section 3, our estimated ranges, seasonal and spatial differences of BrC and its contributions are likely robust. These are the main findings in Section 3.

2. The reviewer's statement is not completely accurate as it neglects the application to the aethalometer data presented in Section 4. This analysis includes some of the key scientific findings of the study.

3. The method we developed remains valuable to the BrC research community even if current high quality multi-wavelength absorption data is limited. As we replied to Reviewer#4, any future absorption measurements at three wavelengths with one in the near-UV and two at longer wavelengths in the visible spectrum can be immediately used to advance the investigation of BrC.

**From the e-mail of co-editor:**

The reviewers though raised significant concerns, which, although are indeed addressed, they are not addressed satisfactorily for every point. You mentioned in one of your replies, that you added "extensive caveats to the text to reflect concerns raised about this issue", and "we include AERONET observations in our analysis despite data limitations, in order to provide at least a qualitative global constraint on BrC". This means that despite you did the best you could, the caveats are still very large, which is something that has been demonstrated in the past. Don't get me wrong, I fully respect your work and its validity, and truly know that you worked hard on this and did the best you could, given the available data. My point is that what the reviewers convinced me for is that your step forward is somewhat to the side (different approach, same result), not just forward. They also made me wonder if the approach is a dead end, and new data are urgently required to really answer the BrC question satisfactorily.

During my initial decision of rejection, I encouraged you to collaborate with someone from AERONET to create a custom retrieval, based on your findings. I understand and respect your reply that this is probably outside the scope of the initial work as planned, but after having received comments from 4 reviewers, the scientific significance of your manuscript received 1 good, 2 fair, and 1 poor. In my opinion this is largely because your work is based on similar assumptions (with large uncertainties) with studies in the past, although you convincingly demonstrated that you have some differences from those past approaches, although not dramatic. I am very well aware that some studies have been published that demonstrated the uncertainties of that approach, and already provided qualitative BrC estimates; adding one more to those is good, but not very original. If, however, you can repeat the AERONET retrievals with your approach in mind, I am sure you will get a truly original manuscript that can make a difference, as I already mentioned in the past.

We want to clarify that our approach is neither built from AERONET nor built for AERONET, though we include AERONET data in the analysis. This is the critical difference between previous studies and our own. We agree that "Some studies have been published that demonstrated the uncertainties of that approach, and already provided qualitative BrC estimates". However, as we discussed in Section 2, those approaches need the information from AERONET and can only be applied within a given dataset similar to AERONET. In contrast, our method is "dataset-free" and "wavelengths-free". Thus the methodology alone provides a significant advance over previous AAE methods.

This flexibility of our method makes it possible to investigate more than only the qualitative BrC absorption, for example, the AAE of BrC, relationship between BrC-MAC and BC/OC ratio, and the aging effect. To the best of our knowledge, there is no previous study providing the global or regional AAE for BrC estimated from multiple measurements. Our analysis for the correlation between BrC-

MAC and BC/OC confirms the study by Saleh et al., 2014 in the laboratory. This has never been investigated from field measurement before, though several modeling studies already use such assumptions (Saleh et al., 2015; Wang Q. et al., 2016). In addition, the flexibility of our method allows it to be applied to aethalometer data to explore the hot topic of the aging effect on BrC absorption. There are a limited number of studies involving the aging of biomass burning plumes (only one from ambient measurements – Forrister et al., 2015) but none that contrast the observed aging of both urban and biomass burning air masses as we do in this manuscript. We believe all of these analyses are "truly original" and a "step forward" for the research of BrC.

We do not see the value in "repeating the AERONET retrievals with our approach" as suggested by the Editor. The AERONET data we are using are AOD and SSA, which are retrieved by solving the radiative transfer equation for a plane-parallel atmosphere. This is purely a radiation calculation which does not include any assumptions for aerosols. Other AERONET retrieval products (which we do not use in the manuscript) do require such assumptions, and perhaps the Editor was under the impression that these were used in our analysis. We have added a sentence to the top of Section 3.1 to explicitly state that this is not the case. Given that we use only the AOD and SSA products for AERONET, we do not see how we could improve upon the AERONET retrieval teams' generation of these products.

[revised manuscript text omitted]

---

## Author Response (AR4)

**Response to editor:**

After carefully considering the arguments raised in the course of the peer review process as well as the adjustments and clarifications implemented in the latest version of the manuscript, I am ready to accept the manuscript for publication in ACP ubject to additional adjustments as detailed below.

5 Please distill the most important messages/statements from the text you added at the end of the manuscript (copied below) into a few sentences to be added at the end of the abstract:

"However, these results are subject to uncertainties associated both with the methodology and with the dataset to which it is applied. The core issue for all methods that use the AAE to estimate the absorption from BrC is the uncertainty associated with the absorption of BC. Our method improves upon previous studies using this approach by using the information from

10 the wavelength dependent measurements themselves and by allowing for an atmospherically-relevant range of BC properties, rather than fixing these at a single assumed value. Additional constraints on BC optical properties and mixing state would help further improve the method.

Given the large uncertainties associated with AERONET retrievals of AAOD, the most challenging aspect of our analysis is that an accurate globally distributed multiple-wavelengths aerosol absorption measurement data set is unavailable at present.

15 Thus, while our study provides qualitative global constraints, and insight into BrC aging processes from the aethalometer observations, achieving a better understanding of the properties, evolution, and impacts of global BrC will rely on the future deployment of accurate multiple-wavelength absorption measurements to which AAE-methods, such as the approach developed here, can be applied."

The entire text appears rather long for being added to the abstract, but on the other hand, there is no hard limit on the length

20 of manuscript abstracts in ACP. Thus, please transfer most if not all of the information from this text into the abstract, especially a clear statement about the limited reliability of the presented results and how more reliable results may be obtained in future studies building on the presented method.

Moreover, please note that in the current PDF version of your manuscript most figures are of low quality (low resolution, and sometimes low contrast/large overlap of labels, lines and symbols). Please improve the figure quality upon revision and

25 make sure that high quality is obtained and maintained in the course of proofreading and final publications.

We thank the Editor for their re-review of this manuscript and their suggestions. We have added several sentences (the majority of the suggested text) to our abstract to convey two key points: (1) how our method differs from previous AAE methods and (2) that our method may be applied to future observations to obtain better quantitative constraints on the global distribution of BrC. We have also improved the quality of the figures as requested.

[revised manuscript text omitted]